# Chemical and genomic characterization of a potential probiotic treatment for stony coral tissue loss disease

Blake Ushijima [1,2 ✉], Sarath P. Gunasekera[2], Julie L. Meyer [3], Jessica Tittl[3], Kelly A. Pitts [2], Sharon Thompson[2,6], Jennifer M. Sneed[2], Yousong Ding [4], Manyun Chen[4], L. Jay Houk[2], Greta S. Aeby [2], Claudia C. Häse[5] & Valerie J. Paul [2 ✉]

Considered one of the most devastating coral disease outbreaks in history, stony coral tissue loss disease (SCTLD) is currently spreading throughout Florida's coral reefs and the greater Caribbean. SCTLD affects at least two dozen different coral species and has been implicated in extensive losses of coral cover. Here we show *Pseudoalteromonas* sp. strain McH1-7 has broad-spectrum antibacterial activity against SCTLD-associated bacterial isolates. Chemical analyses indicated McH1-7 produces at least two potential antibacterials, korormicin and tetrabromopyrrole, while genomic analysis identified the genes potentially encoding an L-amino acid oxidase and multiple antibacterial metalloproteases (pseudoalterins). During laboratory trials, McH1-7 arrested or slowed disease progression on 68.2% of diseased *Montastraea cavernosa* fragments treated ($n = 22$), and it prevented disease transmission by 100% ($n = 12$). McH1-7 is the most chemically characterized coral probiotic that is an effective prophylactic and direct treatment for the destructive SCTLD as well as a potential alternative to antibiotic use.

[1] Department of Biology & Marine Biology, University of North Carolina Wilmington, Wilmington, NC 28403, USA. [2] Smithsonian Marine Station at Fort Pierce, Fort Piece, FL 34949, USA. [3] Department of Soil, Water, and Ecosystem Sciences Department, University of Florida, Gainesville, FL 32611, USA. [4] Department of Medicinal Chemistry, University of Florida, Gainesville, FL 32611, USA. [5] Carlson College of Veterinary Medicine, Oregon State University, Corvallis, OR 97331, USA. [6] Present address: Department of Microbiology and Cell Science, University of Florida, Gainesville, FL 32611, USA. ✉email: ushijimab@uncw.edu; paul@si.edu

Over 20 species of corals along *Florida's coral reef* have been affected by an ongoing mass mortality event resulting in significant losses of various coral species (1–5). These mortalities are attributed to a waterborne disease with unknown etiology called stony coral tissue loss disease (SCTLD), which has been continuously spreading since its first observation in 2014[1]. This outbreak of SCTLD represents one of the many "local" threats Florida's coral reefs face that must be mitigated to prevent catastrophic losses driven by global threats like climate change[2]. The SCTLD outbreak has already negatively impacted coral reef ecosystems in Florida[3,4], so additional local or global threats only further push these reefs past recovery. Unfortunately, by 2018, SCTLD had reached reefs around Mexico[5], and as of December 2022, at least 26 different Caribbean countries/territories are reporting signs of this disease (https://www.agrra.org/coral-disease-outbreak/). The distribution, host range, and persistence of SCTLD make it an unprecedented threat to corals[1,6]. The etiological agent(s) responsible for SCTLD remains unknown; however, evidence suggests some pathogenic bacteria are involved with the disease process because various classes of antibiotics are effective at stopping lesion progression[7]. There are specific groups of bacteria enriched in diseased corals but causation has yet to be established[8,9]. The known pathogen of corals and other invertebrates, *Vibrio coralliilyticus*, is associated with some of the more virulent SCTLD lesions, suggesting that bacterial coinfections may be occurring in addition to a primary malady[10].

Due to the previous studies with antibiotics implicating pathogenic bacteria, the current mitigation strategy for SCTLD uses the broad-spectrum antibiotic amoxicillin mixed with a proprietary paste, Base 2B, applied to in situ corals[11–13]. As of 2021, treatments with amoxicillin in Florida have had a success rate of 67–90% depending on the coral species[11–13]. This effort has potentially saved thousands of coral colonies from imminent death and is currently the only effective field treatment for SCTLD[11–13]. As of November 2022, there have been over 22,000 coral colonies treated with the amoxicillin paste along Florida's Coral Reef, which is still the predominant intervention treatment currently applied (https://www.arcgis.com/apps/dashboards/55a759f02f3c486eb1d29a95f80fba0a). However, the use of antibiotics in the marine environment has raised concerns about contributing to the current problem with the development and spread of antibiotic resistance[14], especially since amoxicillin is commonly used in human and veterinary medicine[15]. The rise of antibiotic-resistant pathogens could render the sole treatment for SCTLD useless and leave managers without a direct mitigation tool. In addition, like most antibiotics, amoxicillin does not provide lasting protection and needs to be repeatedly administered if the initial treatment fails or if the coral is reinfected, which has been observed with SCTLD[11–13]. The latter is concerning, because the agents responsible for SCTLD are unidentified and waterborne;[7] therefore, they are present in the environment and capable of reinfecting hosts. Cumulatively, these concerns highlight the need for additional treatment or prevention strategies for SCTLD. One such alternative is the use of beneficial microorganisms, e.g., probiotics, as a potential treatment for coral diseases[16].

Probiotics could be an improvement over antibiotics because they can be integrated into the host microbiome and act as a prophylactic and/or longer-term treatment (reviewed in refs. [17,18]). Accordingly, probiotics are being studied for various systems including aquaculture, human and veterinary medicine, and corals because of their potential to treat disease and prevent infections. In corals, various microbes have been isolated from healthy colonies that inhibit the growth of pathogenic organisms during in vitro assays[19,20]. In addition, disruption of the healthy coral microflora

has been correlated with reduced host fitness or susceptibility to infection[21,22], which indicates a protective role for these microbes. For example, when the predatory bacterivore *Halobacteriovorax* was inoculated onto the coral *Montastraea cavernosa*, the treatment appeared to prevent the dysbiosis associated with exposure to the coral pathogen *V. coralliilyticus*[23]. In separate studies, treatment with a mixture of bacterial strains reduced the effects of heat stress and a bacterial pathogen on corals[16,24], which may be achieved through a modulation of the coral immune response[16]. While beneficial coral-associated bacteria are understudied in comparison to fields like the human gut microbiome and associated probiotics[25,26], these studies suggest that probiotic bacteria are a viable option for coral disease mitigation.

Studies of potential coral probiotics have so far focused on in vitro experiments, without thorough characterization of the antibacterial potential of the selected bacterial strains[17]. In addition, only one study has applied probiotics directly to corals with a known coral pathogen strain to evaluate bleaching mitigation[16]. Therefore, the study presented here on *Pseudoalteromonas* sp. strain McH1-7 aimed to take a multi-disciplinary approach to understanding the different aspects of this potential coral probiotic. Aquarium studies with live corals determined that McH1-7 can treat SCTLD directly and act as a prophylactic. After establishing the beneficial antibacterial properties of this strain, we used chemical analysis to identify antibacterial small molecules along with genomic analysis to identify putative biosynthetic gene clusters associated with antibacterial compounds. The identification of the korormicin A (hereafter referred to as korormicin in this text) biosynthetic gene cluster provided a target for molecular tools to track McH1-7 on corals over time, which can be used for further endeavors. Altogether, this work describes a potential probiotic treatment for an infectious coral disease as well as one of the best-characterized coral probiotics in terms of its antimicrobial chemistry and effects on a tissue loss disease.

## Results

**Antibacterial-producing microbes isolated from disease-resistant *M. cavernosa*.** To isolate potential probiotics from disease-resistant corals, tissue samples from four *M. cavernosa* fragments used in transmission experiments (two diseased and two disease resistants)[7] were plated onto SWA and TCBS agar, a general marine medium and one selective for *Vibrio* spp., respectively. Using plate counts, all fragments were calculated to have ~$10^5$ CFU/ml of sample plated onto SWA but ranged from $1.60 \times 10^3$ to $1.02 \times 10^5$ CFU/ml for TCBS agar with no correlation to healthy state, suggesting a consistent level of culturable bacteria with variable *Vibrio* populations (Supplementary Table S1). For each fragment, we obtained 111 isolates from the SWA media and then screened for inhibitory activity against three target strains previously isolated from SCTLD lesions (*Alteromonas* sp. McT4-15, *Vibrio coralliilyticus* OfT6-21, and *Leisingera* sp. McT4-56)[10]. Clear trends were observed between the fragment's health state and the number of culturable isolates with antibacterial activity (Supplementary Table S1). For McH1 and McH4 fragments that were disease-resistant, 67 of the 111 (60.36%) and 16 of 111 (14.41%) isolates had antibacterial activity, respectively (Supplementary Table S1). In contrast, fragments McH2 and McH3 that became infected had 1 of 111 (0.90%) and 0 of 111 isolates with inhibitory activity. For the fragment McH1, 46.8% of the isolates with antibacterial activity were circular, smooth, yellow colonies, which was not observed with the other fragments. The isolate with the largest zone of inhibition on all target strains, McH1-7, was selected for further study.

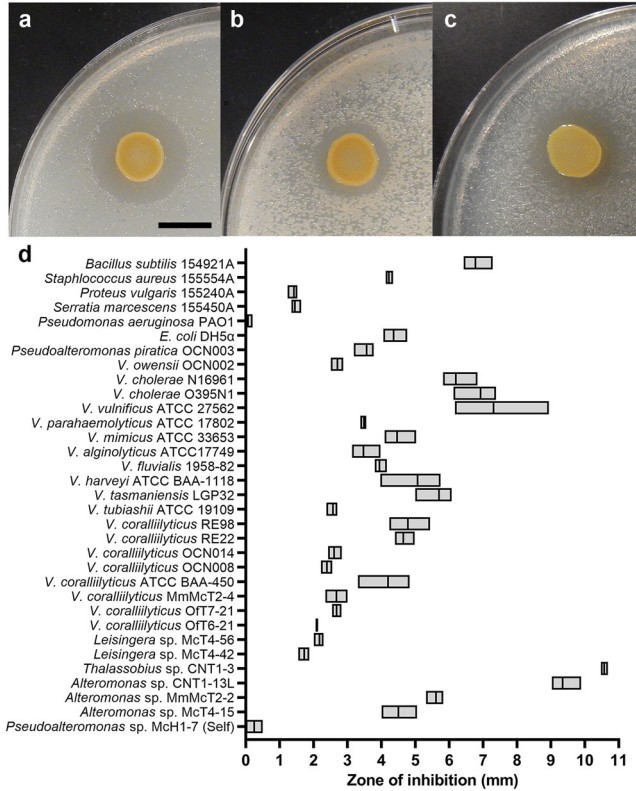

**Fig. 1 Inhibitory activity of isolate McH1-7.** Zone of inhibition (ZOI) of McH1-7 spotted onto target strains **a** *Alteromonas* sp. McT4-15, **b** *Leisingera* sp. McT4-56, or **c** *Vibrio coralliilyticus* OfT6-21. The black scale bar represents 10 mm. **d** Boxplot of ZOI measurements of the radius in mm of McH1-7 on different target strains. The boxes represent the range for three replicates and the middle line indicates the median.

**Isolate McH1-7 has broad-spectrum antibacterial activity.** Isolate McH1-7 was able to inhibit all three target strains isolated from the disease lesions of corals with SCTLD to some degree in disk diffusion assays (Fig. 1a–c). The three target strains, McT4-15, McT4-56, and OfT6-21 had zones of inhibition (ZOIs) between 2.08 and 4.50 mm, with *V. coralliilyticus* OfT6-21 having the smallest ZOI (average 2.08 ± 0.02 mm) and *Alteromonas* sp. McT4-15 having the largest ZOI (average 4.50 ± 0.30 mm) out of the three targets (Fig. 1d). McH1-7 created a ZOI on 31 of the 32 additional target bacteria (Fig. 1d); *Pseudomonas aeruginosa* PAO1 was the only non-inhibited target. Overall, McH1-7 appeared to have broad-spectrum activity across various marine, terrestrial, and human-associated targets as well as both Gram-positive and Gram-negative targets (Fig. 1d). Interestingly, McH1-7 appeared to also have some autotoxic activity against itself (average 0.25 ± 0.14 mm), but this ZOI was less than the other susceptible targets and a ZOI was not observed in every replicate. Together, this suggests that McH1-7 produces a diffusible broad-spectrum antibacterial compound(s).

The organic extract of McH1-7 cells was also active against the three target strains with ZOIs ranging from 0.58 to 5.38 mm. The smallest ZOI was for *Leisingera* sp. McT5-56 (0.58 ± 0.11 mm). The second largest ZOI was *V. coralliilyticus* OfT6-21 (3.71 ± 0.33 mm) and the largest ZOI was *Alteromonas* sp. McT4-15 (5.38 ± 0.13 mm). These results suggested that the McH1-7 antibacterial activity was extractable with ethyl acetate and methanol (1:1); however, the differences to the live-cell assays suggest that some antibacterial compounds were not extractable with the method used or activity was altered for select target bacteria.

**Isolation and characterization of korormicin from McH1-7.** Bioassay-guided isolation using the disk diffusion assays led to one pure compound **1** that displayed the antimicrobial activity in the extract of McH1-7, which gave the molecular formula of $C_{25}H_{39}NO_5$ (HRESI/APCIMS; $m/z$ 456.2813 $[M + Na]^+$; calcd for $C_{25}H_{39}NO_5Na$, 456.2801). The interpretation of DQF COSY, edited HSQC and HMBC experiments indicated the $^1H$ and $^{13}C$ NMR signals of the compound were assignable to three methyls, five olefinic protons, three oxymethines and two ester or amide carbonyl groups. Dereplication using the molecular formula of $C_{25}H_{39}NO_5$ in the marine natural products database MarinLit[27] (http://pubs.rsc.org/marinlit/), together with the described $^1H$ and $^{13}C$ NMR spectral analyses (Supplementary Tables S2 and S3), identified the isolate as korormicin (also called korormicin A) (Supplementary Figs. S1 and S2), an antimicrobial compound previously reported in the literature[28]. The $^1H$ NMR data of compound **1** in DMSO-d$_6$ and the $^1H$ NMR data of korormicin reported in DMSO-d$_6$ (Supplementary Tables S2 and S3) were identical except for minor chemical shift changes observed for the 3H, NH and OH proton signals. These NMR data, together with the comparison of observed specific rotation data to the literature reported specific rotation data $[\alpha]_D^{25}$ −21.4 ($c$ 0.2, MeOH),[28] $[\alpha]_D^{26}$ −24.4 ($c$ 0.29, EtOH), confirmed the structure of the isolated compound as the known korormicin. Complete characterization of korormicin **1** and the $^1H$ NMR, $^{13}C$ NMR, COSY, HSQC, HMBC, and NOESY spectra are given in the supplementary sections (Supplementary Figs. S1–S9). We further identified korormicin directly in the crude extract by comparing its retention time and mass spectrum with those of the standard in the LC-HRMS analysis (Supplementary Fig. S10a, b). In addition to korormicin, we observed multiple korormicin analogs in the extracted ion chromatogram traces, but their intensities were 5 to 100 times lower than korormicin (Supplementary Fig. S11a, b).

**Genomic analysis of McH1-7 identified genes for the antibacterial marinocine and chemical cue tetrabromopyrrole.** The assembled genome of *Pseudoalteromonas* sp. McH1-7 contained 59 contigs over 1000 bp in length. The longest contig was 313,663 bp and the total length of assembled contigs was 5,134,549 bp. The McH1-7 genome had 4429 genes and a G + C content of 42.5%. Of these putative genes, roughly 75% could be assigned to at least one Cluster of Orthologous Genes (Supplementary Table S6). The genome quality is excellent, with an estimated 100% completeness and 1.9% contamination, as assessed by MiGA Online. Raw sequencing reads and the assembled and annotated genome (JABWPQ000000000) are available through NCBI Bioproject PRJNA639770. The annotated genome is also publicly available through the Joint Genome Institute's Integrated Microbial Genomes and Microbiomes database under IMG Genome ID # 2881214049. A thorough comparative genomic analysis of McH1-7 with other marine *Pseudoalteromonas* is ongoing. Here, we focus on the genes and biosynthetic gene clusters that putatively encode the antimicrobial compounds that presumably make McH1-7 an effective probiotic strain.

Fourteen biosynthetic gene clusters were identified in McH1-7 by antiSMASH, including six hybrid clusters with both non-ribosomal peptide synthetases (NRPSs) and type I polyketide synthases (T1PKSs), seven others were NRPS or NRPS-like (likely incompletely assembled), and two were bacteriocins. In addition to the biosynthetic gene clusters identified through antiSMASH, the genes predicted to produce marinocine, including *lodA*, coding for lysine oxidase (IMG GeneID 2881214952) and *lodB*, coding for dehydrogenase flavoprotein (IMG GeneID

2881214951) were identified through homology to the known cluster[29]. A putative tetrabromopyrrole (TBP) gene cluster (IMG GeneIDs 2881214253–2881214258) was located through similarity to the previously published gene cluster in *Pseudoalteromonas* sp. PS5 (GenBank Accession #KR011923.1)[30]. Indeed, we identified TBP in the extracts by comparing its retention time and characteristic isotope MS pattern with those of the standard in the LC-HRMS analysis using the negative ion mode (Supplementary Fig. S12a, b). Overall, the genomes of *Pseudoalteromonas* sp. McH1-7 and *Pseudoalteromonas* sp. PS5 had an average nucleotide identity of 98.91% in shared genes. The gene clusters for marinocine and TBP were 97.9% and 98.7% similar between strains McH1-7 and PS5, respectively.

The genome of McH1-7 also contains two genes that encode putative peptidoglycan DD-metalloprotease family proteins that were identified through similarity to the well-characterized pseudoalterin genes in the marine *Pseudoalteromonas* strain CF6-2 (GenBank Accession #HQ005379.1). Pseudoalterin degrades peptidoglycan and likely has higher antimicrobial activity against Gram-positive bacteria[31]. The first pseudoalterin-like metalloprotease gene in McH1-7 (IMG GeneID 2881218238) is 73% similar to the gene in strain CF6-2, and the second pseudoalterin-like metalloprotease gene in McH1-7 (IMG GeneID 2881216045) is less than 51% similar to the gene in strain CF6-2. Both genes are annotated as belonging to COG0739 for murein DD-endopeptidase MepM/murein hydrolase activator NlpD and the gene products of both are annotated as belonging to the M23 family metallopeptidases (pfam01551). In addition, McH1-7 has homologs of all twelve core genes of the Type II Secretion System needed to export pseudoalterins[31], with the general secretion proteins encoded by *gspCDEFGHIJKLMN* (IMG GeneIDs 2881216855–2881216866).

**A previously unknown biosynthetic gene cluster for korormicin was putatively identified.** A potential *kor* gene cluster was identified from the sequenced genome of *Pseudoalteromonas* McH1-7, and an almost identical gene cluster was present in the genomes of *Pseudoalteromonas* sp. PS5, J010, and *P. peptidolytica* DSM 14001 (Supplementary Note and Supplementary Figs. S13 and S14). Of note, *Pseudoalteromonas* sp. J010 is known to produce multiple korormicin analogs[32], and we also detected multiple korormicin analogs in the crude extract of McH1-7 by LC-HRMS analysis (Supplementary Fig. S11). In McH1-7, the 52-kb *kor* cluster encodes 36 open-reading frames (ORFs) (Table 1), whereas the 10-kb region upstream of the cluster encodes potential primary pathway proteins that might not directly contribute to the biosynthesis of korormicins. An operon of seven putative vitamin B$_{12}$ biosynthetic genes is located immediately downstream of the *kor* cluster and this operon is not considered as part of the cluster because this operon is not associated with the korormicin biosynthetic genes in *Pseudoalteromonas* sp. J010. On the other hand, vitamin B$_{12}$ may regulate the metabolism of ubiquinone[33], a key component of the korormicin cellular target[34] and makes for an intriguing interplay between these compounds. A detailed description of the putative korormicin gene cluster and biosynthesis can be found in the Supplementary Methods and depicted in Fig. 2 and Table 1.

**McH1-7 can slow or arrest lesion progression on diseased *Montastraea cavernosa*.** Diseased *M. cavernosa* fragments were treated in aquaria with McH1-7 to test its efficacy as a direct treatment for SCTLD. Overall, mortality and disease progression rates were lowered when diseased fragments were treated with McH1-7 compared to controls (Fig. 3 and Supplementary Table S4). For the control coral fragments (Fig. 3a–d), 59.1% had

complete mortality while the remaining fragments had progressive lesions over 21 d ($n = 22$). In contrast, for fragments treated with McH1-7 (Fig. 3e–h) 59.1% had disease progression arrested, 9.1% had progressive tissue loss, and 31.8% had complete mortality ($n = 22$). The standardized disease progression rates were significantly slower with fragments treated with McH1-7 (mixed-effects ANOVA; treatment $P = 0.020$, time $P < 0.0001$, interaction $P = 0.083$; $n = 22$) (Fig. 3I). Likewise, the area under the curves (AUC) of tissue loss progression was higher for fragments treated with McH1-7, which suggests treatment slowed tissue loss progression (paired $t$ test, $P = 0.002$, $n = 22$) (Fig. 3j).

For the diseased *M. cavernosa* colonies used in the above-mentioned experiments, 18 colonies were opportunistically screened for the toxic metalloprotease VcpA, nine of which were positive. Previous studies found that the presence of VcpA correlated with increased rate of disease progression[10]. This correlation between the presence of VcpA and greater mortality was again observed with the *M. cavernosa* fragments used in this current study (Fig. 4). AUC analysis of the disease progression rates showed that their VcpA status (two-way ANOVA, $P = 0.005$, $n = 9$) and treatment (two-way ANOVA, $P = 0.029$, $n = 9$) did have an effect, and there was no significant interaction (two-way ANOVA, $P = 0.523$, $n = 9$) (Fig. 4b). Overall, these results suggested that McH1-7 may be a viable treatment for SCTLD based on these aquarium trials, but further investigation is warranted because *V. coralliilyticus* coinfections may diminish treatment success.

**McH1-7 protects *M. cavernosa* from SCTLD while the beneficial effects may be transmissible.** One strength of probiotics is their potential as a prophylactic treatment; therefore, healthy *M. cavernosa* were treated with McH1-7 to evaluate its ability to protect against SCTLD transmission. Overall, McH1-7 was able to protect pre-treated *M. cavernosa* fragments in contact with diseased fragments over a 21-day period after a single treatment (Fig. 5 and Supplementary Table S5). There was no obvious damage from intraspecific competition between control corals (Fig. 5a–d), so any lesions observed with the experimental corals were attributed to disease transmission. The non-treated healthy fragments in contact with diseased fragments (Fig. 5e–h) had a disease transmission rate of 33.3% ($n = 12$) within 4–6 days of contact, which was similar to previous transmission studies[7]. In contrast, none of the fragments pre-treated with McH1-7 (Fig. 5i–l), developed any disease signs during the 21-day experiment (log-rank (Mantel–Cox) test, $P = 0.032$, $n = 12$). Surprisingly, the diseased corals in contact with pre-treated fragments had slower disease progression compared to the diseased corals that were in contact with non-treated fragments (mixed-effects ANOVA; treatment $P = 0.004$, time $P < 0.001$, interaction $P < 0.0001$; $n = 12$) (Fig. 5m). The area under the curves (AUC) of tissue loss progression was higher for fragments pre-treated with McH1-7 (Paired $t$ test, $P = 0.0001$, $n = 12$) (Fig. 5n). These data suggest that a single treatment with McH1-7 can protect *M. cavernosa* from SCTLD for at least 21 days and the beneficial effects may be transmissible to diseased corals in aquaria.

**Detection of McH1-7 in healthy corals over time after inoculation.** The presence of McH1-7 on coral fragments were tracked over time using gene copies of the Kor23 gene of the putative korormicin biosynthetic gene cluster. This gene was detected by ddPCR in most aquaria water samples, including samples from 1 h after inoculation, and 1 day, 3 days, 7 days, 21 days, and 28 days post inoculation (Fig. 6 and Supplementary Fig. S15). These results are described in more detail in the supplementary

**Table 1 Features of the gene products in the predicted korormicin (kor) clusters of *Pseudoalteromonas* McH1-7.**

| Protein | Size[a] | Homolog and origin | ID/SI[b] | Putative function |
|---|---|---|---|---|
| Kor1 | 426 | RXF06809.1, *Pseudoalteromonas* sp. PS5 | 99/99 | Class III transaminase |
| Kor2 | 338 | RXF06808.1, *Pseudoalteromonas* sp. PS5 | 100/100 | Aspartate carbamoyltransferase |
| Kor3 | 144 | RXF06807.1, *Pseudoalteromonas* sp. PS5 | 100/100 | Uncharacterized membrane protein |
| Kor4 | 560 | RXF06806.1, *Pseudoalteromonas* sp. PS5 | 100/100 | CLC voltage-gated chloride channel |
| Kor5 | 112 | RXF06805.1, *Pseudoalteromonas* sp. PS5 | 100/100 | Iron–sulfur cluster insertion protein ErpA |
| Kor6 | 58 | WP_164488155.1, *P. rubra* | 79/91 | Hypothetical protein |
| Kor7 | 82 | RXF06804.1, *Pseudoalteromonas* sp. PS5 | 100/100 | Hypothetical protein |
| Kor8 | 80 | RXF06803.1, *Pseudoalteromonas* sp. PS5 | 100/100 | Helix–turn–helix transcriptional regulator |
| Kor9 | 461 | RRS10230.1, *Pseudoalteromonas* sp. J010 | 99/100 | MFS transporter |
| Kor10 | 205 | RRS10229.1, *Pseudoalteromonas* sp. J010 | 100/100 | Peptidylprolyl isomerase |
| Kor11 | 185 | NLR13385.1, *P. peptidolytica* | 99/99 | Uncharacterized lipoprotein |
| Kor12 | 376 | RXF06799.1, *Pseudoalteromonas* sp. PS5 | 99/99 | rRNA methyltransferase |
| Kor13 | 197 | RXF06798.1, *Pseudoalteromonas* sp. PS5 | 98/98 | α-ketoglutarate-dependent dioxygenase AlkB |
| Kor14 | 104 | RXF06797.1, *Pseudoalteromonas* sp. PS5 | 100/100 | BolA family transcriptional regulator |
| Kor15 | 396 | RXF06796.1, *Pseudoalteromonas* sp. PS5 | 100/100 | Trans-2-enoyl-CoA reductase family protein |
| Kor16 | 206 | NLR13390.1, *P. peptidolytica* | 99/99 | 3-isopropylmalate dehydratase small subunits |
| Kor17 | 467 | NLR13391.1, *P. peptidolytica* | 99/99 | 3-isopropylmalate dehydratase large subunit |
| Kor18 | 381 | NLR13392.1, *P. peptidolytica* | 99/99 | 3-isopropylmalate dehydrogenase |
| Kor19 | 210 | NLR13393.1, *P. peptidolytica* | 100/100 | α-ketoglutarate-dependent dioxygenase |
| Kor20 | 465 | NLR13394.1, *P. peptidolytica* | 99/99 | 3-isopropylmalate dehydratase large subunit |
| Kor21 | 513 | RXF06790.1, *Pseudoalteromonas* sp. PS5 | 99/99 | 2-isopropylmalate synthase |
| Kor22 | 1255 | NLR13396.1, *P. peptidolytica* | 99/99 | FAAL-DH*-T |
| Kor23 | 1483 | NLR13397.1, *P. peptidolytica* | 99/99 | PKS (KS-AT*-KR-T) |
| Kor24 | 1819 | NLR13398.1, *P. peptidolytica* | 99/99 | PKS (KS-AT-DH-KR-T) |
| Kor25 | 1434 | NLR13399.1, *P. peptidolytica* | 99/99 | PKS (KS-AT*-KR-T) |
| Kor26 | 1378 | NLR13340.1, *P. peptidolytica* | 99/99 | NRPS (C-A-T-TE) |
| Kor27 | 335 | NLR13341.1, *P. peptidolytica* | 100/100 | Fatty acid desaturase |
| Kor28 | 451 | RXF06783.1, *Pseudoalteromonas* sp. PS5 | 99/99 | NADH:ubiquinone reductase subunit A |
| Kor29 | 400 | RXF06782.1, *Pseudoalteromonas* sp. PS5 | 100/100 | NADH:ubiquinone reductase subunit B |
| Kor30 | 249 | RXF06781.1, *Pseudoalteromonas* sp. PS5 | 99/100 | NADH:ubiquinone reductase subunit C |
| Kor31 | 210 | RXF06780.1, *Pseudoalteromonas* sp. PS5 | 100/100 | NADH:ubiquinone reductase subunit D |
| Kor32 | 202 | RXF06779.1, *Pseudoalteromonas* sp. PS5 | 100/100 | NADH:ubiquinone reductase subunit E |
| Kor33 | 408 | RXF06778.1, *Pseudoalteromonas* sp. PS5 | 100/100 | NADH:ubiquinone reductase subunit F |
| Kor34 | 343 | RXF06777.1, *Pseudoalteromonas* sp. PS5 | 99/99 | FAD:protein FMN transferase |
| Kor35 | 80 | RRS10205.1, *Pseudoalteromonas* sp. J010 | 98/98 | DUF1496 domain-containing protein |
| Kor36 | 70 | WP_010372193.1, *Pseudoalteromonas* | 100/100 | $(Na_+)$-NQR maturation NqrM |

[a]Amino acid; [b]identities/similarities (%).

text. Briefly, ddPCR results suggest that this probiotic is not maintained at relatively high levels on *M. cavernosa* over time as well as not dominating the coral microflora (Supplementary Note and Supplementary Fig. S16). In all, these results suggest the beneficial effects of McH1-7 on healthy corals is not due to it outcompeting and dominating the coral microflora nor does it require a concentrated density of this strain, which may be a positive trait for a probiotic treatment.

## Discussion

This study describes the discovery and application of *Pseudoalteromonas* sp. McH1-7, a bacterial probiotic effective against a coral tissue loss disease and the deepest characterization of its chemistry and application. This strain is effective as a direct treatment for SCTLD, and perhaps more importantly, as a pre-treatment of healthy coral fragments to protect against disease transmission. Although the mechanisms responsible for the protective attributes of McH1-7 are uncertain, its broad-spectrum antibacterial activity is thought to be involved because broad-spectrum antibiotics are effective against SCTLD[11–13]. This strain is predicted to produce at least four different types of antibacterial compounds, which include the antibiotic korormicin. Although korormicin has been previously isolated, characterized, and utilized in other studies investigating membrane bioenergetics of human pathogens like *Vibrio cholerae*[34,35], we have

described the putative biosynthetic pathway for this antibiotic and suggest its potential role in a probiotic treatment.

Korormicin was first isolated from the marine bacterium *Pseudoalteromonas* sp. strain F-420, which was isolated from the surface of the alga *Halimeda* sp. collected from Palau[28]. Subsequent studies on *Pseudoalteromonas* sp. F-420 collected from the same location reported five more related korormicins[36]. Total synthetic studies by two groups have confirmed the stereochemistry of 4 *S*,3'*R*,9'*S*,10'*R* for korormicin[37,38]. Another study of *Pseudoalteromonas* strain J010, isolated from the surface of the crustose coralline alga *Neogoniolithon fosliei*, yielded thirteen natural products[39]. These natural products included another set of five new korormicins in addition to the six previously known korormicins[36]. Even though various analogs have been identified and this compound has been used for important membrane bioenergetics studies[34,35], the biosynthetic genes had not been identified. The identification of a potential biosynthetic pathway in this study opens a tremendous number of possibilities for this compound. For example, it provided a relatively unique target for ddPCR to track McH1-7 on corals, a valuable tool for follow up work. In addition, more in-depth studies on the mechanism of McH1-7 are planned, which would include elucidating the roles that compounds like korormicin play in this strain's probiotic activity. However, the roles of the putative biosynthesis genes must be further confirmed with additional mutagenesis and

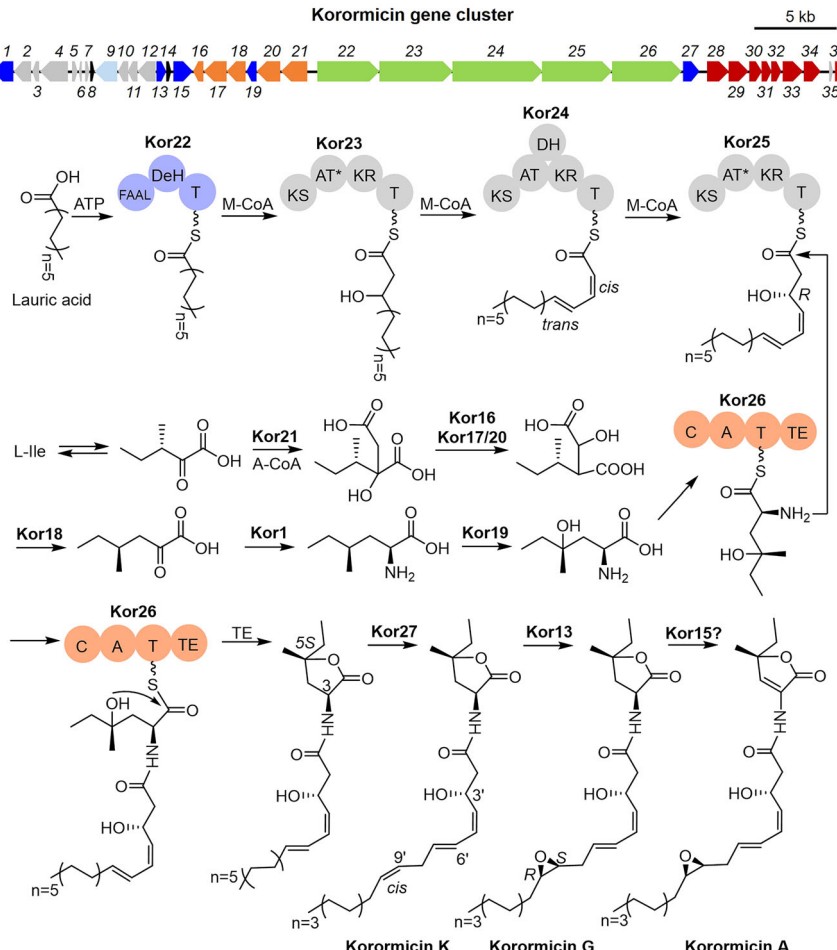

**Fig. 2 The putative biosynthetic pathway of korormicins.** The 52-kb *kor* gene cluster was identified from the genome of *Pseudoalteromonas* sp. McH1-7. The PKSs and NRPSs are shown in green, while the genes involved in the formation of the nonproteinogenic amino acid (2 *S*)-2-NH₂-4-OH-4-methylhexanoate are in orange. Genes in the operon of NADH:ubiquinone reductase assembly (NQR) are presented in dark red, while those in dark blue may encode modification enzymes. Two transcriptional regulators are in black, and the rest are in gray. The proposed pathway is supported by the predicted functions of gene products in the identified *kor* gene cluster.

genetic complementation, and the specific functions of the translated enzymes have yet to be confirmed.

In addition to korormicin and minor amounts of korormicin analogs, McH1-7 is predicted to produce an L-amino acid oxidase (L-AAO) related to the known enzyme marinocine[29,40] due to the presence of a *lodA* (lysine oxidase) and *lodB* (dehydrogenase flavoprotein) homologs present in its genome. L-AAOs catalyze the deamination of specific L-amino acids and release ammonium and hydrogen peroxide, which is responsible for their broad-spectrum antimicrobial activity[29,40]. There are various marinocine-like compounds like AlpP from *P. tunicata* D2 or PfaP from *P. flavipulchra* JG1[41,42], and these different types will utilize different L-amino acid substrates (63–65). However, this conversion of amino acids to hydrogen peroxide potentially contributes to their autotoxic properties[41–45].

McH1-7 was also found to produce the small molecule 2,3,4,5-tetrabromopyrrole (TBP) by LC-HRMS analysis in comparison to a TBP standard. This compound is an intermediate for the bio-synthesis of the pyrrole antibiotic pentabromopseudilin[46], how-ever, some bacteria lack *bmp5-8* which results in the production of TBP[30,47]. Extracts of TBP were reported to have antibacterial activity[48] as well as antialgal activity hypothesized to be via the rapid induction of ROS and the release of Ca²⁺ stores[49]. TBP is also a settlement cue for larvae of multiple coral species[32,50], suggesting a multifunctional role for this compound and

additional uses for McH1-7. Interestingly, korormicin, L-AAOs, and TBP are each predicted to induce oxidative stress directly or indirectly in target bacteria, which may be the reason for the broad-spectrum antibacterial activity of McH1-7. However, it is currently unclear how the activities of these three antibacterial compounds are related (e.g., if they have additive or synergistic activity) and if they are directly responsible for the effectiveness of McH1-7 against SCTLD. The success of the current SCTLD field treatment with the antibiotic amoxicillin[11–13], does suggest the antimicrobial compounds produced by McH1-7 play a role in its protective attributes.

In contrast to the ROS-related antibacterial compounds, McH1-7 is predicted to produce bactericidal metalloproteases (pseudoalter-ins) through in silico analysis of its genome. A pseudoalterin recently described in *Pseudoalteromonas* sp. CF6-2 appeared to kill Gram-positive isolates via degradation of the peptide chains linking the glycan subunits making up the cell wall (peptidoglycan)[31]. The CF6-2 pseudoalterin did not appear to affect Gram-negative iso-lates, which they partially attributed to glycine, an activator of expression for the metalloprotease that is only present in Gram-positive peptidoglycan crosslinks. It is also possible that Gram-negative bacteria are protected by the lipopolysaccharide layer surrounding their cell wall or by protective enzymes in their peri-plasmic space. However, it is currently unknown if these anti-bacterial metalloproteases are active in McH1-7 or the extent of

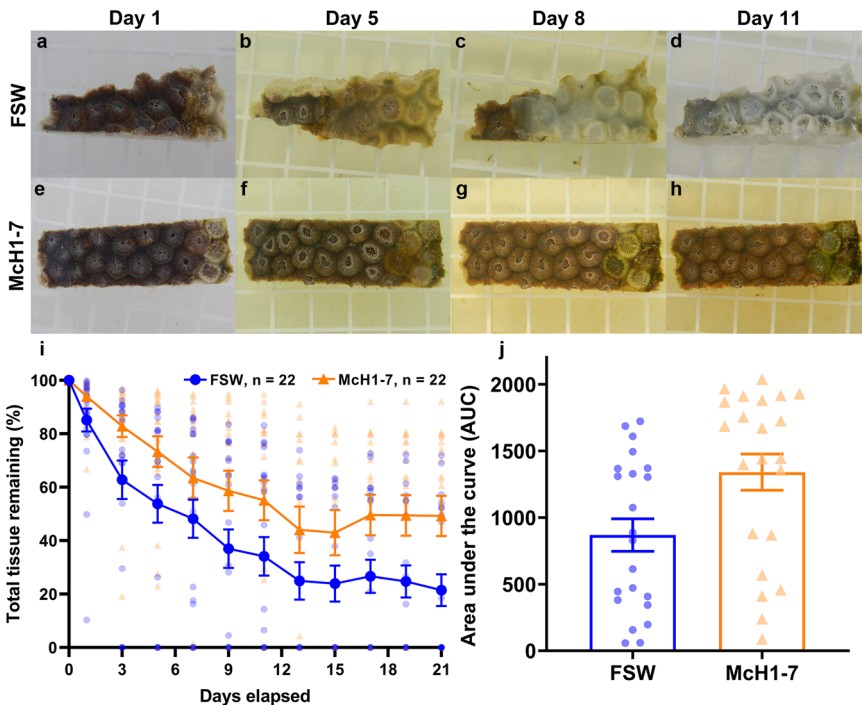

**Fig. 3 Direct treatment of SCTLD with McH1-7. a–h** Representative photos of diseased *M. cavernosa* treated with filtered seawater (FSW) (**a–d**) or McH1-7 (**e–h**). Photos represent days 1, 5, 8, and 11 of the experiment. In this example, the FSW treatment represents 100% tissue loss after 11 days (**d**), while the McH1-7 treatment represents 5% tissue loss after 11 days (panel **h**). **i** Average percentage of total tissue remaining on *M. cavernosa* colonies treated with FSW (blue circles) or McH1-7 (orange triangles) over time (mixed-effects model ANOVA; treatment $P = 0.020$, time $P < 0.0001$, interaction $P = 0.083$; $n = 22$). The error bars represent the standard error of the mean. **j** Area under the curve (AUC) calculations for the diseased *M. cavernosa* treated with FSW or McH1-7 (paired $t$ test, $P = 0.002$, $n = 22$ each treatment).

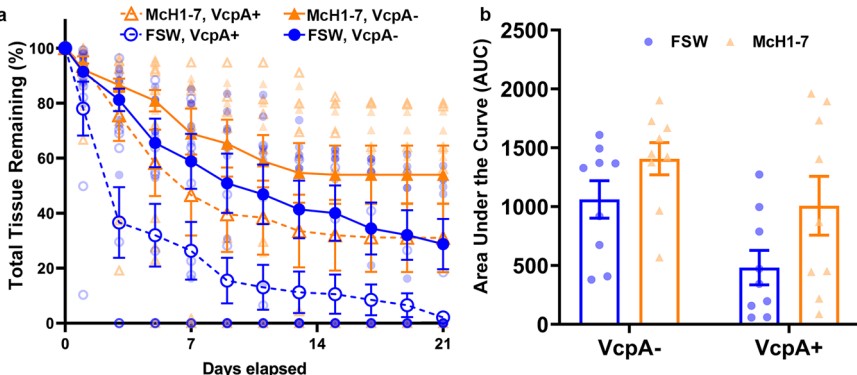

**Fig. 4 The relationship between VcpA assay result and McH1-7 efficacy. a** Average percent of total tissue remaining on diseased *M. cavernosa* that are VcpA$^-$ and exposed to just FSW (blue solid line with closed blue circles) or McH1-7 (orange solid line with closed orange triangles) as well as was VcpA$^+$ fragments exposed to FSW (blue dashed line with open blue circles) or McH1-7 (orange dashed line with open orange triangles). **b** Area under the curve (AUC) calculations for percent total tissue remaining on the diseased *M. cavernosa* that are VcpA$^-$ and exposed to just FSW (blue bar) or McH1-7 (orange bar) as well as VcpA$^+$ fragments exposed to FSW (blue bar) or McH1-7 (orange bar). Two-way ANOVA of the disease progression rates showed that VcpA status ($P = 0.005$, $n = 9$) and treatment ($P = 0.029$, $n = 9$) did have an effect, and there was no significant interaction ($P = 0.523$, $n = 9$). The error bars represent the standard error of the mean.

their activity, but it does present an important avenue for future research. In all, McH1-7 is predicted to produce at least four different antibacterial compounds that affect different molecular targets, which would theoretically reduce the risk of pathogens evolving resistance and increase the range of microbes affected. These attributes would be beneficial for a SCTLD treatment since the primary agent is unidentified and additional opportunistic bacterial infections may be occurring[10].

While recent work has demonstrated that the addition of beneficial bacteria can alter coral microbiome composition[24,51,52],

we found that an effective probiotic may not need to comprise a major proportion of the host microbiome to confer beneficial effects, and this is consistent with other recent attempts at coral microbiome manipulation. For example, protection from heat stress was conferred by inoculation with a cocktail of six beneficial bacterial strains, yet only half of the strains could be detected, and those that were detected were present at less than 1% relative abundance[16]. Similarly, the microbial communities of coral larvae were successfully manipulated with the addition of a cocktail of seven probiotic strains, though most were detected at

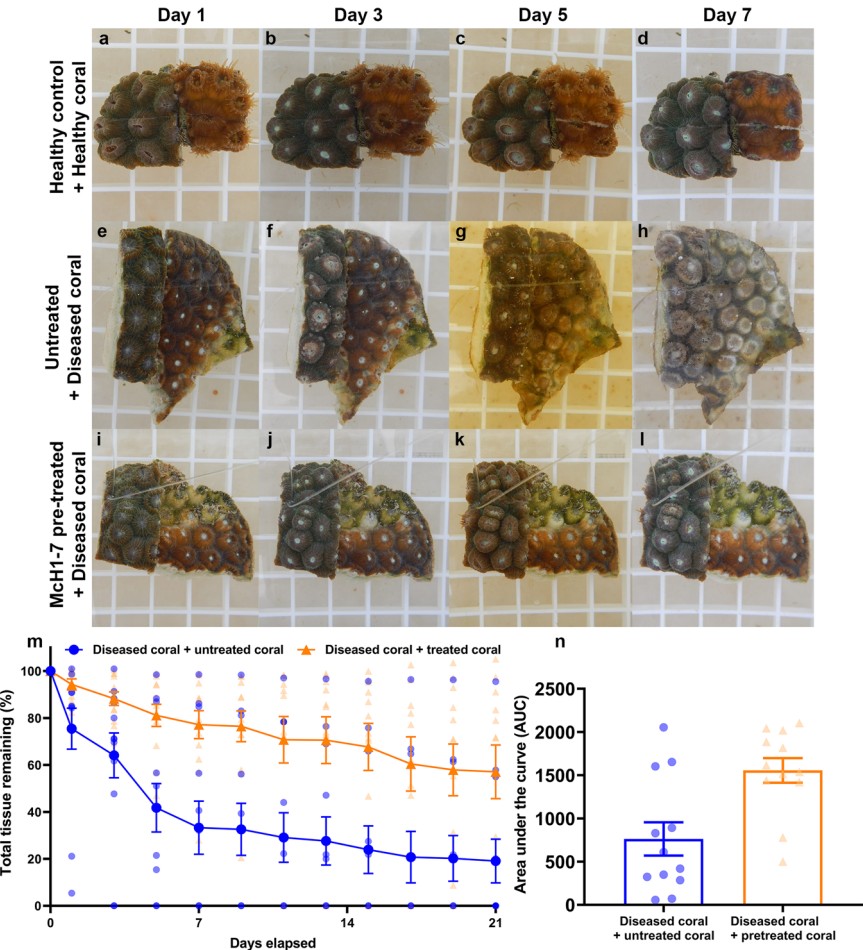

**Fig. 5 Protection from disease transmission with McH1-7. a–d** Photos of control *M. cavernosa* consisting of a healthy fragment pre-treated with McH1-7 (left) in contact with an untreated healthy fragment (right), **e–h** untreated healthy fragment (left) in contact with a diseased fragment (right), and **i–l)** a healthy fragment pre-treated with McH1-7 (left) in contact with a diseased fragment (right). Photos represent days 1, 3, 5, and 7 of the experiment. **m** Average percentage of total tissue remaining on diseased *M. cavernosa* colonies in contact with untreated healthy fragments (blue circles) or McH1-7 treated healthy fragments (orange triangles) over time (mixed-effects model ANOVA; treatment $P = 0.004$, time $P < 0.001$, interaction $P < 0.0001$; $n = 12$). **n** Area under the curve (AUC) calculations for the diseased *M. cavernosa* in contact with untreated fragments (blue bar) or fragments pre-treated with McH1-7 (orange bar) (paired $t$ test, $P = 0.0001$, $n = 12$). The error bars represent the standard error of the mean.

≤5% relative abundance[53]. Here, McH1-7 does not appear to dominate the host microflora after treatments (Supplementary Fig. S16) yet is able to protect against SCTLD transmission as well as potentially benefit nearby diseased corals (Fig. 5). This suggests that being able to significantly outcompete the host microbiome may not be a required, or even a desirable trait for an effective probiotic, as it could be speculated that overdominance of the host microbiome would inhibit adaptability due to a loss of the communities' functional diversity. One can speculate upon the mechanisms for this process, but this study demonstrates how a potentially effective prophylactic treatment for coral disease does not necessarily have to be the dominant microbe on a coral.

This discovery of McH1-7 is both timely and incredibly important as SCTLD is an unprecedented emerging threat to all reefs in the Caribbean. Although there is currently an effective treatment for SCTLD, the broad-spectrum antibiotic amoxicillin[11–13], there are concerns about the potential widespread use of this antibiotic contributing to the pollution of marine environments[15]. Furthermore, the reliance on a single antibiotic will inevitably result in antibiotic-resistant pathogens, which can occur through various mechanisms[54,55]. In addition, amoxicillin, as with other antibiotics, may not be a feasible prophylactic treatment due to the breakdown of the drug[56] and

potential concerns for disrupting the normal microflora present on the corals. For example, a previous study found that treating *Oculina patagonica* with the broad-spectrum antibiotic nalidixic acid allowed the pathogen *Vibrio shiloi* to infect this coral previously thought to have evolved resistance to this bacterium[21]. Further, the results in this study suggest that, at least partially, resistance to SCTLD involves acquiring specific microflora constituents like McH1-7, which could be removed through the treatment of healthy corals with broad-spectrum antibiotics.

McH1-7 represents an entirely new avenue of field treatments for SCTLD. Due diligence was taken to ensure the safety of this strain as well as establish pipelines for evaluating future coral probiotics. This included, most importantly, testing this strain on corals to ensure safety, but also included steps taken to identify the antimicrobial compounds produced by this bacterium. While genome sequencing and analysis was able to identify the genes for the L-AAO and was important for identifying the putative *kor* gene cluster, it was in combination with the described chemical analysis that allowed for a more complete characterization. In all, this discovery is an important step forward into the development of novel treatments for coral disease, while also providing a general workflow for the characterization of additional beneficial strains for corals.

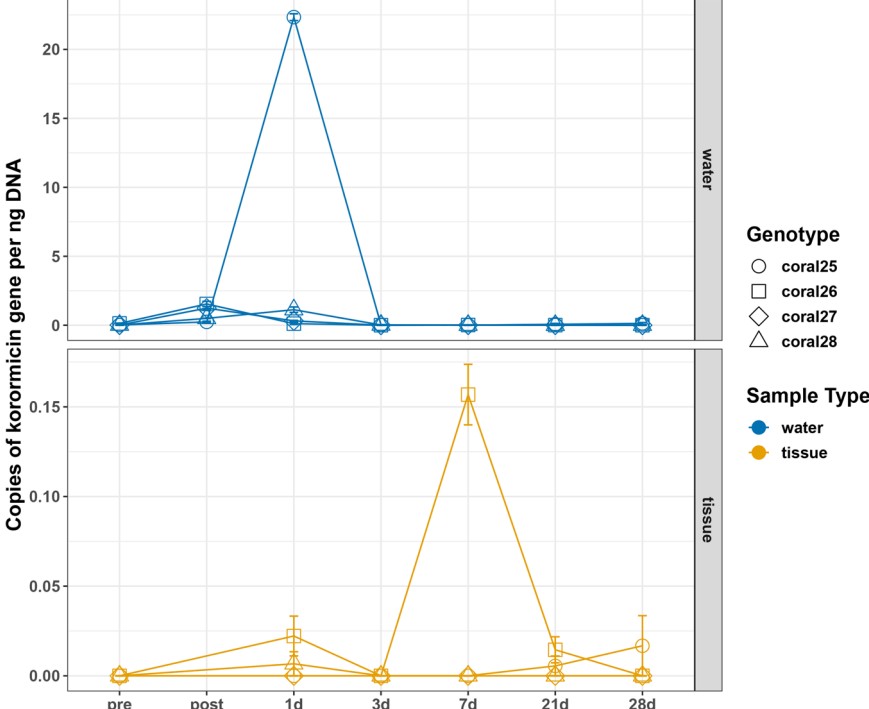

**Fig. 6 Detection of korormicin gene copies over time after inoculation with Pseudoalteromonas McH1-7.** Points show the mean copy number per ng of DNA of the Kor23 gene from triplicate reactions of ddPCR and the standard error is shown by the error bars. "Pre" indicates samples taken before inoculation with the probiotic strain McH1-7. "Post" indicates samples taken one hour after inoculation in water samples only. Each *M. cavernosa* genotype was cut into six fragments prior to the experiment and one fragment was sacrificed at each time for genetic analysis. Note the difference in the y-axis scales, as higher copy numbers were detected in the water compared to tissue samples.

## Methods

**Coral collection and husbandry.** Filtered oceanic seawater was used for all experiments, coral husbandry, and bacterial growth media, which was collected and prepared in an identical manner described in a previous study[10]. Briefly, ocean seawater was successively filtered through 20, 1.0, 0.5, and 0.35-μm pore filters into storage containers. The seawater in storage was kept under constant recirculation through a 20-μm pore filter, a filter canister with ROX 0.8 aquarium carbon (Bulk Reef Supply), and a 36-watt Turbo-twist 12× UV sterilizer (Coralife) in series. Immediately before use, the seawater was filtered through a 0.22-μm pore filter and is referred to as filtered seawater (FSW) throughout the text.

Portions of diseased colonies were collected while scuba diving using a hammer and chisel from various locations around Broward County and the Florida Keys in compliance with the appropriate permits from the Florida Fish and Wildlife Conservation Commission and Florida Keys National Marine Sanctuary, respectively. A detailed list of the coral colony IDs and their collection locations are detailed in Supplementary Tables S4 and S5. To reduce the chance of collecting corals with the same genotype, colonies at least 5 m apart were selected for use. All healthy corals used in the experiments described here were received from the NOAA Key West coral nursery.

All corals collected from the field or from the nursery were transported to the Smithsonian Marine Station (SMS) facility in Fort Pierce, FL after being wrapped in plastic bubble wrap moistened with seawater and then placed in a cooler. Upon arrival at the SMS facility, each fragment was gently rinsed with FSW to remove excess mucus and loose debris. Healthy and diseased corals were handled and maintained in separate areas of the facility[10]. Briefly, all corals were maintained in temperature-controlled systems and were initially maintained at the same temperature as the collection site but were then slowly adjusted to 28 °C (a maximum 0.5 °C change per day) before use in experiments. Diseased corals were held in buckets with 13 L of FSW and a weighted airline in outdoor systems under ambient light conditions with a clear plastic canopy and one layer of shade cloth that blocked ~50% of the sunlight. Partial water changes (~50% water exchange) were conducted at different intervals depending on the experiment (see below). Sterilized plastic scoops were used for all water changes. Scoops were rinsed and scrubbed in a 10% calcium hypochlorite solution, rinsed with freshwater several times, and then left to dry for at least 24 h before each use. Apparently healthy corals were kept in an indoor facility within multiple large recirculating water table systems each holding ~570 L of FSW. A row of six blue-white 30 cm² LED panels (HQPR) were fixed above each table providing 150–250 μmol photons m⁻² s⁻¹ for captive corals. Targeted feeding of healthy corals occurred three times weekly alternating between a mix of Reef Roids (PolypLab), Marine Snow (Two Little Fishes), and LPS or SPS Max (Dr. G's Marine Aquaculture) according to each

manufacturer's recommendations. Corals were not fed during experiments to minimize water fouling with diseased corals.

All corals were fragmented with a masonry saw (Husqvarna MS 360) fitted with a 35.56 cm diamond, continuous rim, circular saw blade. To cool the blade during cutting, FSW was flowed over it using a flow-through system, so the FSW was not recirculated. The corals were covered with plastic bubble wrap during cutting to prevent tissue damage. The saw blade was cleaned with 70% ethanol and the entire system was flushed with freshwater and left to completely dry for at least 24 h between cutting sessions. Diseased corals were used within 24–72 h of fragmentation due to the progression of tissue loss.

**Bacterial growth, specimen sampling, and inhibition assays.** All bacteria used in this study are listed in Supplementary Table S6. Marine bacteria were grown in seawater broth (SWB) that consists of 2 g/L yeast extract, 4 g/L tryptone mixed with 1 L of FSW, with 15 g/L of agar added prior to autoclaving for solid media (seawater agar (SWA))[10]. For any cultures grown up for coral experiments, the SWB was supplemented with 2 ml/L of sterile glycerol (GSWB). All cultures were grown at 28 °C with constant shaking at 150–200 rpm for liquid cultures, unless otherwise specified.

During SCTLD transmission experiments from a previous study[7], a group of four healthy *M. cavernosa* colonies (Supplementary Table S1) from the Key West nursery were repeatedly exposed to diseased *M. cavernosa* collected from Broward County. Two of the initially healthy fragments developed disease (McH2 and McH3), but two fragments (McH1 and McH4) remained healthy. Tissue and mucus samples from each of these four fragments (two diseased and two non-diseased) were collected using a sterile, needless syringe by gently agitating the surface of the coral while in FSW and plated onto SWA and TCBS agar. Counts of CFU (colony forming units)/ml were determined for each media type. The two healthy corals were further sampled for bacterial isolation. Syringes were slowly drawn up while agitating the coral to collect mucus and small fragments of tissue. The sample was then expelled into a sterile conical tube and subsequently vortexed for 2–3 min. After mixing, the sample was serially diluted in autoclaved FSW and then plated onto SWA. After incubation for 48–72 h colonies were picked based on morphology to increase isolate diversity and streaked out onto SWA before incubation. After growth, isolates went through 1–2 more rounds of streaking to ensure purity before growth in SWB and cryopreservation at −80 °C with 30% glycerol (final volume).

Laboratory screens for isolates that produce antibacterial compounds were conducted on SWA using stationary phase liquid cultures grown for 24 h. A drop culture assay was used by spotting 10 μl of the culture tested for antibacterial

activity (tester strain) onto an SWA plate spread with 50 µl of an isolate that would have their growth inhibited (target strain). The target strains used for the inhibition assays were *Vibrio coralliilyticus* strain OfT6-21(GenBank genome sequence JABSNA000000000), *Alteromonas* sp. strain McT4-15 (GenBank genome sequence JAJALG000000000), and *Leisingera* sp. strain McT4-56 (GenBank genome sequence JAJALF000000000), which were previously isolated from *Orbicella faveolata* and *Montastraea cavernosa* coral fragments displaying signs of SCTLD[10]. After one target strain was spread onto a plate, the tester strains were then spotted, allowed to dry, and then incubated for 24 h. Each screen was done in triplicate with each of the three target strains. After incubation, the zone of inhibition (ZOI) was measured using a caliper from the edge of the target strain spot directly to the edge of growth inhibition (if present). This inhibition bioassay was used to identify bacterial strains that inhibited the growth of putative pathogenic strains.

**Chemical extraction and bioassay-guided fractionation**. McH1-7 isolated from healthy *M. cavernosa* coral fragment McH1 was grown on 25 plates of SWA for 48 h, scraped from the surface of the plates and lyophilized to produce sufficient biomass for chemical studies (0.527 g). The freeze-dried biomass was extracted with a mixture of ethyl acetate and methanol (1:1) to give 0.162 g of the lipophilic extract. The extract and all subsequent fractions were tested in disk diffusion assays (Kirby-Bauer tests)[57] to utilize bioassay-guided fractionation to identify the anti-microbial component(s) of the extract. Target strains OfT6-21, McT4-15, and McT4-56 were used for disk diffusion assays to track the bioactivity through the isolation process. Crude extracts were solubilized in methanol at 25 mg/mL and dosed onto 6 mm sterile paper disks at 5.0 µL (125 µg), in triplicate, while fractions of the extracts were tested at their respective proportional concentrations. Disks were placed under the laminar flow hood to allow evaporation of the solvent, and then disks were placed onto the surface of the inoculated agar. A methanol disk and plain paper disk were used as negative controls. Each target was seeded after 24 h growth in 2 ml SWA cultures, which equated to approximately $10^9$ CFU/ml, onto SWA plates (50 µL of liquid cultures spread with sterile glass beads). Plates with pathogens and disks were incubated for 24 h at 28 °C and were scored by measuring zones of inhibition (ZOIs) radius around the paper disks.

The bioactive extract (0.134 g) was chromatographed on a column of C-18 silica using a step gradient system of 20% $H_2O$-MeOH, 10% $H_2O$-MeOH and MeOH to give three sub-fractions (F1–F3). The active fraction F2 (7 mg) was further purified by reversed-phase HPLC (semi-prep 250 × 10 mm, 5 µm, RP-18, flow 3.0 mL/min) using 25% $H_2O$-MeOH to give a pure, antimicrobial compound as a colorless oil (2.4 mg, $t_R$ = 40.1 min, yield, 2.1% extract). Additional details can be found in the Supplementary Methods.

The crude extract of McH1-7 grown in seawater broth in January 2022, which was pelleted, lyophilized, and extracted with 2:2:1 EtOAc: MeOH: water, was analyzed by LC-HRMS. LC-HRMS and HRMS/MS experiments were conducted on Thermo Scientific™ Q Exactive Focus mass spectrometer with Dionex™ Ultimate™ RSLC 3000 uHPLC system, equipped with H-ESI II probe on Ion Max API Source. Water with 0.1% formic acid (A) and acetonitrile with 0.1% formic acid (B) were used as the mobile phases to separate analytes on a Phenomenex Kinetex® C18 column (1.7 µm, 2.1 × 50 mm, 100 Å). A typical LC program with a 0.6 ml/min flow rate included 5% B for 2 min, 5–95% B in 13 min, 95% B for 2 min, 95 to 5% B in 0.5 min, and re-equilibration in 5% B for 2 min. The eluents from the first 2 min and last 3 min were diverted to a waste bottle by a diverting valve. MS1 signals were acquired under the Full MS-positive ion mode or negative ion mode covering a mass range of $m/z$ 300–1500, with a resolution at 35,000 and an AGC target at 1e6. Standards of purified korormicin and synthetic tetrabromopyrrole (TBP) were included in the analyses to confirm the presence of these compounds.

Gas chromatography-mass spectrometry (GC-MS) was also used to detect the presence of TBP, a known biosynthetic product of some *Pseudoalteromonas* bacteria[30], in the extract of McH1-7. Lyophilized cells were extracted in 50:50 methanol/ethyl acetate. The crude extract was fractionated using silica column chromatography. The most nonpolar fraction (75:25 hexanes/ethyl acetate) was analyzed by GC-MS (Shimadzu GC-17A, GC-MS-QP5000) using a HP-5MS column (30 m × 0 25 mm, Agilent Technologies, Inc). The presence of TBP was confirmed by comparing the resulting mass spectrum to the known mass spectrum of TBP.

**Genome sequencing and analysis**. Genomic DNA was extracted from *Pseudoalteromonas* sp. McH1-7 cells pelleted from 1.5 ml of turbid cultures in saltwater broth using a DNeasy PowerSoil Kit (Qiagen). Sequencing libraries were prepared with a Nextera Flex kit (Illumina) and sequenced on an Illumina MiSeq with the 2 × 150 bp v. Two cycle at the University of Florida Interdisciplinary Center for Biotechnology Research. Raw sequencing reads were quality-filtered with the Minoche[58] filtering pipeline in illumina-utils v. 2.3[59] and Illumina adapters and Nextera transposase sequences were removed with cutadapt v. 1.8.1[60]. Quality-filtered reads were assembled with SPAdes v. 3.13.0[61]. Biosynthetic gene clusters were identified with the online antiSMASH database bacterial version 5.1.2[62]. Genome quality was assessed with the Microbial Genomes Atlas (MiGA) online[63]. Average nucleotide identity between *Pseudoalteromonas* sp. McH1-7 and *Pseudoalteromonas* sp. PS5[30] was assessed with the Average Nucleotide Identity calculator from the enveomics toolbox[64]. Custom BLAST searches were conducted

from the command line with NCBI BLAST + v. 2.10.1[65]. The korormicin (*kor*) gene cluster was analyzed by locally installed antiSMASH 5[62] to predict ORFs. The predicted ORFs were functionally annotated by performing BLASTp[66] against the NCBI database. For the phylogenetic analysis, the enzymes were aligned in MEGAX by ClustalW method and the phylogenetic tree was created using the Maximum-likelihood method with 1000 bootstrap replications[67].

**Treatment of diseased corals with McH1-7**. To evaluate the effect of McH1-7 on SCTLD, diseased corals ($n = 22$) were cut into pieces that were treated with a course of the putative probiotic in comparison with untreated controls. Diseased corals were cut with a masonry saw (described above) to approximately split the disease lesion and the remaining living tissue between both fragments. The fragments were then gently rinsed with FSW before being individually placed into clear aquaria with 5 L of FSW. Each aquarium held only one coral fragment. For each experimental replicate (control and treated coral), all fragments originated from the same colony and shared the same contiguous disease lesion. These tanks were held in the SMS outdoor systems (described above) and water temperature was maintained at 28 °C. Water changes of ~50% of the total tank volume took place every other day using sterilized scoops. After each partial water change, the corals were re-inoculated with McH1-7 unless otherwise stated.

For McH1-7 inoculations, a culture of McH1-7 was revived from a frozen glycerol stock culture by streaking onto SWA and incubating overnight. After incubation, 2–3 colonies from the plate were inoculated into fresh SWB and incubated for ~15 h. The liquid cultures were then diluted 1:100 in fresh SWB and incubated with shaking until reaching an optical density measured at 600 nm ($OD_{600 nm}$) between 0.8 and 1.0. This $OD_{600 nm}$ is approximately equivalent to $10^9$ CFU/ml according to plate counts. 50 ml aliquots of the cultures were centrifuged at $8000 \times g$ for 5 min to pellet the cells, and the media was decanted off. For inoculation, all air bubblers were turned off and 1–2 ml of tank water was used to gently resuspend the bacterial pellet with a transfer pipette. The resuspended culture was then pipetted gently over the fragment. The air bubblers were left off for 2 h post inoculation. There was a total volume of 5 L of FSW in each tank, so the concentration of inoculated bacteria was approximately $10^7$ CFU/ml of tank water. This process was repeated after each partial water change for a total of 10 inoculations over a period of 21 days. Data and statistical analyses are described below.

**Transmission experiments with McH1-7**. To test if McH1-7 treatments could prevent disease transmission, three treatment tanks were used per experimental block ($n = 12$ each treatment). First, the disease control (DC) tank consisted of a diseased *M. cavernosa* fragment in contact with an apparently healthy *M. cavernosa* fragment. Second, the experimental (EXP) tank had a diseased fragment (same genotype as the DC tank) in contact with a healthy fragment (same genotype as the DC tank) pre-treated with McH1-7. Third, the healthy control (HC) tank had an apparently healthy fragment (same genotype as the DC and EXP fragment) pre-treated with McH1-7 in contact with another healthy fragment from another colony. The fragments treated with McH1-7 for the EXP and HC tanks were inoculated once and left for 48 h in a separate tank in an identical manner as described above for treatment of diseased corals, while the healthy fragments for the DC control tanks were treated the same except without McH1-7 inoculation. After McH1-7 inoculation at ~$10^7$ CFU/ml of tank water, the fragments were left in the tanks for 24 h before a partial water change and then left in the same tanks for another 24 h. The fragments were then moved into their respective tanks with new FSW and a diseased fragment (the DC and EXP tanks) or a healthy fragment (the HC tanks). All fragments had been initially cut to approximately the same height (2 cm) so when they were tied together using small gauge monofilament fishing line, they had some tissue contact. Both fragments were placed side by side and left upright in direct contact with each other. Transmission experiment fragments were not inoculated again with McH1-7, but had partial water changes every other day. The transmission fragments were photographed every other day for 21 days, and analysis was conducted in a similar manner as the direct treatment experiment (see below).

**Detection of McH1-7 in healthy corals over time after inoculation through droplet digital PCR (ddPCR)**. Four genotypes of healthy *M. cavernosa* colonies were cut into seven fragments as described above and placed in four 5-L clear aquaria (one per genotype). The fragments were left to recover with daily partial water changed in FSW for 72 h post-fragmentation. Each tank was dosed with McH1-7 as described above. A detailed description of this protocol and analysis of the coral microbiomes is provided in the supplementary text (Supplementary Methods).

**Statistics and reproducibility**. For inhibition assays with McH1-7 against the variety of different target strains (Fig. 1d), cultures of each target were grown up with three biological replicates originating from a single colony that was then assayed with one technical replicate for each. Conclusions were not drawn from the average sizes of the ZOIs created and this data was treated as a simple binary result of having a ZOI or not to demonstrate the range of McH1-7 antibacterial activity.

The ZOI measurements of the radius in mm were represented on a box plot with the median value and range of ZOI measurements.

For all coral experiments, the selection of coral fragments was limited by their availability in the field for diseased corals or availability from nurseries for healthy corals. Coral fragments were grouped by colony origin, so that each experimental replicate used fragments from the same colony, i.e., genetically identical specimens. All statistical analyses and graphs were completed in GraphPad Prism (version 8.4.3) using the built-in functions. A fragment was considered to have complete mortality when no living tissue was left. Fragments that survived the 21-day monitoring period but had progressive lesions were considered to have slow progression. Fragments with disease progression that arrested during the 21-day monitoring period and did not progress any further were considered to have stopped progression. An exception was for lesions that stopped and then began to progress again, which were considered to have slow progression (if they also survived the 21-day period). In addition, all coral fragments were repeatedly photographed every other day and the area of living tissue remaining was measured using ImageJ (U.S. National Institutes of Health). Four genotypes were missing photographs on days 13 and 15 for both control and McH1-7 treated corals. The grating that the fragments sat on was used as the set scale because each square measured 1.5 cm × 1.5 cm. The total area of living tissue on each fragment was measured at the start of the experiment (day 0), 24 h post, and then every other day after that. If a fragment had complete mortality, the photograph from the previous day was measured and included in the analysis. Each measurement in cm$^2$ was divided by the measurement from day 0 and multiplied by 100 to calculate the percent remaining tissue at each time point to standardize the data for the unequal sizes of the coral fragments. The percent of tissue remaining was plotted over time and then the area under the curve (AUC) was calculated for each fragment, with a lower AUC value corresponding to faster disease progression. The AUC values for the control and experimental fragments were compared using a paired t test. The diseased M. cavernosa colonies used were opportunistically screened for the toxic metalloprotease VcpA, which is indicative of V. coralliilyticus coinfections[10]. The effect of the presence or absence of V. coralliilyticus and the interaction this bacterium plays with McH1-7 treatment was analyzed using a two-way ANOVA comparing their AUC data.

**Reporting summary**. Further information on research design is available in the Nature Portfolio Reporting Summary linked to this article.

## Data availability

The sequencing reads for the 16 S rRNA gene amplicons are available in NCBI under BioProject PRJNA801145. The sequencing reads and genome assembly for *Pseudoalteromonas* sp. strain McH1-7 are available in NCBI under BioProject PRJNA639770. All tissue measurement data associated with live coral experiments are publicly available on the Figshare database and can be accessed at https://figshare.com/s/18f60a82d908192c4014[68].

## Code availability

Analysis and plotting of the ddPCR and 16 S rRNA gene amplicon data were performed in R as described at https://github.com/meyermicrobiolab/McH1-7_Probiotics_Trials.

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

## Acknowledgements

This work was supported by funding provided by the Florida Department of Environmental Protection Office of Resilience and Coastal Protection-Southeast Region (V.P., J.M., G.A., and B.U.), the Smithsonian George E. Burch Fellowship to BU, the National Science Foundation (IOS-1728002) (V.P., G.A., J.M., and C.H.), the NOAA Coral Reef Conservation Program Award NA19NOS4820114 (B.U. and V.P.), and NIH R35GM128742 (Y.D.). Field collections were authorized under Florida Fish and Wildlife Conservation Commission Special Activity Licenses SAL-16-1702A-SRP, SAL-17-1702-SRP, SAL-18-1702-SRP, and Florida Keys National Marine Sanctuary permits # FKNMS-2017-128-A2 and FKNMS-2019-160. The authors would like to thank many Smithsonian staff for their assistance, including Woody Lee for his outstanding technical support as well as Olivia Carmack, Yesmarie De La Flor, Benjamin Johnson, Kendall Lee, Maria Estefany Madrigal, Andrea Mason, Jaqueline Reu, Raquel Wetzell, and Kylie Zimmerman for assistance with experimental set up and coral husbandry. We are grateful to Garret M. Rubin (Department of Medicinal Chemistry, University of Florida) for running and analyzing LC/HRMS samples of McH1-7 extracts. We greatly appreciate the synthetic standard of TBP provided by Vinayak Agarwal and Dongqi Yi (School of Chemistry and Biochemistry, Georgia Institute of Technology). We also thank Broward County personnel, Angel Rovira, Ken Banks, Kirk Kilfoyle, and Pat Quinn, as well as Nova Southeastern University personnel Karen Neely, Emily Hower, and Kevin Macaulay for their help with coral collections. We also thank the Harbor Branch Oceanographic Institute at Florida Atlantic University spectroscopy facility for 600 MHz NMR spectrometer time and optical rotation measurement. This is Smithsonian Marine Station contribution #1188.

## Author contributions

The authors confirm their contribution to the paper as follows: study conception and design: B.U., G.S.A., C.C.H., and V.J.P.; data collection: B.U., S.G., J.L.M., J.T., K.P., S.T., M.C., J.S., L.J.H., and V.J.P.; analysis and interpretation of results: B.U., S.G., J.L.M., K.P., Y.D., M.C., and V.J.P.; draft manuscript preparation: B.U., K.P., Y.D., J.M., and V.J.P. All authors reviewed the results and approved the final version of the manuscript.

## Competing interests

The authors declare no competing interests.
