## [Peer Review File · Communications Biology]

Reviewers' comments:

Reviewer #1 (Remarks to the Author):

The manuscript "Chemical and genomic characterization of the first potential probiotic treatment for a coral disease threatening the Caribbean" presents a very timely and necessary approach to mitigate the ongoing devastating impacts caused by SCTLD, through the use of specific coral probiotics for disease.

This is an urgent issue, not only due to the current spread of this threat, but also due to the (dangerous) current use of broad-spectrum antibiotics to cope with the disease. Overall, I think authors could even cite the recent science to policy paper published by the International Coral Reef Society (ICRS) (Knowlton et al., 2021), where a plan to save coral reefs is presented. One of these pillars is mitigation of local pollution, or local impacts. SCTLD is an example of a major "local" problem that is certainly boosted by climate change and, at the same time, increases the losses caused by thermal stress. I believe this interaction with climate change, as well as the role of diseases in making thermal bleaching even worse, should be cited in the introduction, as it highlights (even more) the importance of this work.

Overall the experiments were well conducted and the results are well presented. This work presents an impressive characterization of the antibacterial potential of the probiotic strain and its role on both containing and preventing SCTLD infection. I therefore fully support its publication. I have some minor suggestions that, I hope, can improve the document.

Line 38: involved or correlated? Shifts can be a response to SCTLD but not necessarily a cause (as nicely explained in lines 39-40). It might be safer to say that some pathogenic bacteria are correlated with the disease status or just say that some pathogenic bacteria may be involved with the disease process.

Line 47: Can you add a short sentence about the current status in terms of scale and permits for this type of application in the field (in case you have such information)? I believe the use of probiotics is way more appropriated/safer and the current status on the use of antibiotics reinforces the importance of your work.

Line 60: I would use both references, the one cited and the original one (Peixoto et al., 2017), where the term Beneficial Microorganisms for Corals (BMC) is proposed (to describe probiotics that are specifically selected and tailored for application on corals) and the one you cited (Peixoto et al., 2021), where the concept and BMC mechanisms are revisited and updated. Both are key publications to give you a strong background on the importance and potential of coral probiotics.

Lines 69-70: Reference number 20 has also shown the effects of a mixture of probiotic bacteria on the mitigation of *V. coralliilyticus* infection, which was pretty expressive considering qPCR surveys for *V. coralliilyticus* (and this is a key data considering your topic). You should therefore say: "...reduced the effects of heat stress and disease on corals..."

Line 74: Rosado et al., 2019 presents results that indicate the antagonistic effects of a probiotic consortium on the mitigation of a coral pathogen (*V. coralliilyticus*), on corals. Also, other studies have characterized specific mechanisms that are not covered in this manuscript, like the genetic and physiological potential associated with nitrogen cycle, ROS mitigation, DMSP degradation, etc. Even though, this is for sure the most well characterized coral probiotic and the authors present an impressive characterization of the antibacterial potential of the strain – which can be highlighted here, as well. I therefore suggest you say: "Studies of potential coral probiotics have so far focused on in vitro experiments, without thorough characterization of the antibacterial potential of the selected bacterial strains. In addition, only one of them has actually applied probiotics on corals to evaluate

disease mitigation (20).”, or so.

Figure 3 and other legends: Maybe consider using “Filtered seawater” rather than “FSW” the first time you use it for each legend?

Line 205: I suggest you include the higher % of survivorship rate observed, as your results are quite impressive.

Line 209: 9 out of 18 is 50%, does that mean you could not correlate the presence of VcpA with increased rate of disease? This sentence/result is not clear.

Line 228: This is very promising.

Line 248: I struggle with the term “best characterized”. I see your point and I agree this is such an impressive survey regarding antibacterial traits, but one could argue that there are other traits being described for other BMCs that are not covered here. So, in terms of diversity of traits, this may not be the “best” but it is for sure the “deepest” analysis regarding a specific trait (antibacterial production). Perhaps “as well as the best characterized antibacterial coral probiotic to date.”? Or something around these lines.

Line 255: Perhaps delete “and” in this sentence, so that it reads “...korormicin has been previously isolated, characterized and utilized in other studies...”

Line 317: I am not sure I follow the narrative here. I believe this study is aligned with the thermal stress studies, where most of the probiotics did not outcompete the native microbiome. In those studies, some probiotics were already dominant members of the microbiome – but sensitive to thermal stress – and some were not (and continued to be found in low abundances or weren’t even found, as members of the detectable microbiome). In any case, for all of these studies, there were some discussions that are in agreement with your results and discussion here (which is very positive as it shows consistent results over different coral probiotic applications). For all these studies, it seems that probiotics don’t need to be dominant or even persistent. In some cases, most probiotics won’t even be detected, as they only seem to trigger a beneficial microbiome restructuring/succession (e.g., Santoro et al., 2021). Overall, they do seem to guarantee niche protection and provide beneficial mechanisms that, collectively, protect corals against different impacts. These strategies may be slightly different when comparing disease and thermal stress, as the production of antibacterial compound seem to be crucial for SCTL, for example, although they may all benefit from overall health improvement. Please adapt this paragraph to reflect that all studies developed so far seem to point to the same direction, that even if some of the probiotic players are not dominant or persistent, they still seem to trigger beneficial outcomes, as this actually sounds like excellent news.

Lines 328-322: I agree this is an incredible contribution, and maybe it should be also highlighted in the abstract, if possible.

Figure 6: I am not sure I understand it, have you quantified it in non-inoculated corals, to evaluate the natural occurrence of these genes, as a control?

Line 385: Did you keep healthy and sick corals under different conditions and feeding regimes? For how long? Did you acclimate them all under the same conditions and using the same system before starting the experiment?

Line 490: Please add the number of replicates here.

Line 508: When did you collect your samples for microbiome/ddPCR analyses? Just before inoculation?

Reviewer #2 (Remarks to the Author):

The paper by Ushijima describes the identification of a *Pseudoalteromonas* bacterial strain with novel potential use as a biological treatment for a coral disease. In my opinion, the research described in this paper is quite appealing as it is a good example of the potential use of microorganisms as biological pest control agents, with the additional difficulty to do so in aqueous media. However, as a reviewer I miss additional information about the state of the art in this field in a marine environment.

In addition, as a natural product chemist I believe that a deeper study of the chemical composition of the studied microorganism is necessary, particularly when the authors claimed in the title to have done it.

1.- The authors identified korormicin as one of the responsible agents of the bioactivity. Did they find any related molecule? Many derivatives have been found from the same genera and may be present here as well.

2.- Regarding chemical analysis, the identification of korormicin is trivial as it is a well known molecule. However, it is not easy to confirm this point by the reader. I could not find any figure in the manuscript and it is absolutely necessary to follow the tables in SI. I would recommend to include a table with NMR chemical differences between the herein reported values and the previously described ones (from synthesis).

3.- TBP was also identified but I cannot find any experimental data in the SI section to confirm its identity.

In summary, I think the idea and discoveries behind this paper could be quite useful but I miss more scientific soundness from a chemical point of view. Although the reported research is well done, the identification of two-three well known compounds is not enough to claim the chemical characterization of a microorganism. Alternatively, if korormicin is the main responsible of the reported action I would make it clear although this would clearly lower the novelty of the discovery reported. Thus, taking into account that it seems difficult to use *Pseudoalteromonas* directly as a biological control agent shortly, I clearly miss a deeper chemical study of the bacterial extract or even a metabolomics study.

Reviewer #3 (Remarks to the Author):

This manuscript describes a probiotic and prophylactic bacteria called Mch1-7 for an ongoing coral disease outbreak – stony coral tissue loss disease. The authors use an interdisciplinary approach including microbiology, biochemistry, aquatic experiments, and omics approaches to characterize the efficacy, antibacterial properties, and other characteristics of Mch1-7. The results from the tank experiments are convincing, showing reduced mortality rates and tissue loss from SCTLTD when Mch1-7 is applied before and during infection. I also think the manuscript reads well, and I appreciated the detailed methods.

However, I wondered why the authors did not conduct a more thorough examination of the genome of Mch1-7. The genome was characterized with antiSMASH, and then the other analyses were conducted through a targeted approach looking for specific genes. It seems relevant to conduct an entire functional analysis of the genome with KASS, KEGG, PFAM, or any other protein/functional annotation

method. There may be other key genes/pathways that may bring to light the mechanisms of Mch1-7.

As for the 16S rRNA results, the authors claim that there is no change to the microbiome by Mch1-7 but also that Mch1-7 was only detected in one sample at one time point. Since Mch1-7 was only detected in one sample I would assume that there would be no obvious microbiome changes caused by Mch1-7 and that it of course isn't a dominant member if it can't be detected. Based on these results it's difficult to conclude that Mch1-7 colonized the corals – perhaps it mostly colonized sick corals when added? I think the authors need to provide a deeper discussion on this discrepancy. I recognize that the authors detected the Mch1-7 Kor23 gene in the ddPCR assay in 5/24 samples, but this is still a low number of samples and the ddPCR values themselves are low, with not even one copy detected per ng. What does that mean in terms of the number of cells? Since Mch1-7 is cultured, why not determine its detection threshold with this ddPCR assay. Overall I think this was a well-done study, and I only have a few line-by-line comments below.

Line 32 Describe what is meant by “not fully characterized ...”

Line 43 Antibiotics was used to imply that this was a bacteria disease and not the other way around.

Line 147 What is the number of ORFs and GC content of the genome?

Line 235 I think this needs to be its own section with a header

Line 259 add space after Kromoiacin

Line 285 Let the reader know what method was used to detect this molecule.

Line 299 again remind the reader how this result was generated.

Line 314 - 317 This sounds like an introduction to a discussion and sounds out of place here. Perhaps split the discussion up into sections or rewrite.

Line 325 How does this compare to probiotics used in other systems including corals? Is it typical that the probiotic isn't detected?

Line 440 Which antibiotics were tested for the Kirby-Bauer tests?

Line 461 Why was only the most non-polar fraction selected?

Line 556 I think it's helpful to add that these were healthy corals in the header of this section.

In the supplementary it is difficult to provide comments since line numbers were not provided. But there is a paragraph that is completely underlined and I don't understand why.

Fig 2 The text on the top portion of the figure (the cluster) is low resolution.

Fig 3 Remove the triangles from "I" since there is no key and there are no triangles in "K". In the legend add the percentage of tissue loss that is represented in A-H.

Fig S9 add a Kb scale

Fig S12 A. I think the data would be easier to interpret if each time point was plotted against the distance to the centroid.

B. I think this should be plotted at a lower taxonomic level like genus especially if *Pseudoalteromonas* is apparent on the plot.

Author responses are in blue and line numbers refer to the “marked” word document submitted.

Reviewers' comments:

Reviewer #1 (Remarks to the Author):

The manuscript "Chemical and genomic characterization of the first potential probiotic treatment for a coral disease threatening the Caribbean" presents a very timely and necessary approach to mitigate the ongoing devastating impacts caused by SCTLD, through the use of specific coral probiotics for disease.

This is an urgent issue, not only due to the current spread of this threat, but also due to the (dangerous) current use of broad-spectrum antibiotics to cope with the disease. Overall, I think authors could even cite the recent science to policy paper published by the International Coral Reef Society (ICRS) (Knowlton et al., 2021), where a plan to save coral reefs is presented. One of these pillars is mitigation of local pollution, or local impacts. SCTLD is an example of a major "local" problem that is certainly boosted by climate change and, at the same time, increases the losses caused by thermal stress. I believe this interaction with climate change, as well as the role of diseases in making thermal bleaching even worse, should be cited in the introduction, as it highlights (even more) the importance of this work.

Thank you, this citation was added and better links to your points were included in the introduction (lines 46-49).

Overall the experiments were well conducted and the results are well presented. This work presents an impressive characterization of the antibacterial potential of the probiotic strain and its role on both containing and preventing SCTLD infection. I therefore fully support its publication. I have some minor suggestions that, I hope, can improve the document.

Line 38: involved or correlated? Shifts can be a response to SCTLD but not necessarily a cause (as nicely explained in lines 39-40). It might be safer to say that some pathogenic bacteria are correlated with the disease status or just say that some pathogenic bacteria may be involved with the disease process.

We have modified this sentence according to the reviewers' suggestions and added an explanation and a citation to further clarify the statement (lines 53-54).

Line 47: Can you add a short sentence about the current status in terms of scale and permits for this type of application in the field (in case you have such information)? I believe the use of probiotics is way more appropriated/safer and the current status on the use of antibiotics reinforces the importance of your work.

We do not have the current status of the Caribbean intervention efforts, however, we have include current metrics (as of December 2022) for the Florida response according to the Coral Disease Intervention Dashboard, which we have included a link to in the text (lines 50-52).

Line 60: I would use both references, the one cited and the original one (Peixoto et al., 2017), where the term Beneficial Microorganisms for Corals (BMC) is proposed (to describe probiotics that are specifically selected and tailored for application on corals) and the one you cited (Peixoto et al., 2021), where the concept and BMC mechanisms are revisited and updated. Both are key publications to give you a strong background on the importance and potential of coral probiotics.

We have added in the citation for Peixoto et al., 2017 (lines 89-98).

Lines 69-70: Reference number 20 has also shown the effects of a mixture of probiotic bacteria on the mitigation of *V. coralliilyticus* infection, which was pretty expressive considering qPCR surveys for *V. coralliilyticus* (and this is a key data considering your topic). You should therefore say: "...reduced the effects of heat stress and disease on corals..."

Agreed, we have modified the sentence according to the reviewer's suggestion. However, "a bacterial pathogen" was used instead of "disease" to be a little more accurate (lines 101-102).

Line 74: Rosado et al., 2019 presents results that indicate the antagonistic effects of a probiotic consortium on the mitigation of a coral pathogen (*V. coralliilyticus*), on corals. Also, other studies have characterized specific mechanisms that are not covered in this manuscript, like the genetic and physiological potential associated with nitrogen cycle, ROS mitigation, DMSP degradation, etc. Even though, this is for sure the most well characterized coral probiotic and the authors present an impressive characterization of the antibacterial potential of the strain – which can be highlighted here, as well. I therefore suggest you say: "Studies of potential coral probiotics have so far focused on in vitro experiments, without thorough characterization of the antibacterial potential of the selected bacterial strains. In addition, only one of them has actually applied probiotics on corals to evaluate disease mitigation (20).", or so.

The sentence has been modified using the reviewer's suggestion (lines 106-109).

Figure 3 and other legends: Maybe consider using "Filtered seawater" rather than "FSW" the first time you use it for each legend?

This has been changed in the figure legend (Figure 3).

Line 205: I suggest you include the higher % of survivorship rate observed, as your results are quite impressive.

Excellent point, we have stated the present survival in these sentences (line 279-292).

Line 209: 9 out of 18 is 50%, does that mean you could not correlate the presence of VcpA with increased rate of disease? This sentence/result is not clear.

Thank you for pointing out the confusion. Our observation was that that 50% of the fragments were positive for VcpA. The control fragments that were positive with VcpA still has overall faster disease progression than the control fragments that were negative for VcpA. This was further clarified in the paragraph (line 293-303).

Line 228: This is very promising.

Thank you very much.

Line 248: I struggle with the term “best characterized”. I see your point and I agree this is such an impressive survey regarding antibacterial traits, but one could argue that there are other traits being described for other BMCs that are not covered here. So, in terms of diversity of traits, this may not be the “best” but it is for sure the “deepest” analysis regarding a specific trait (antibacterial production). Perhaps “as well as the best characterized antibacterial coral probiotic to date.”? Or something around these lines.

We have modified the sentence to reflect the reviewer comment (lines 373-375).

Line 255: Perhaps delete “and” in this sentence, so that it reads “...korormicin has been previously isolated, characterized and utilized in other studies...”

The change has been made (line 381-384).

Line 317: I am not sure I follow the narrative here. I believe this study is aligned with the thermal stress studies, where most of the probiotics did not outcompete the native microbiome. In those studies, some probiotics were already dominant members of the microbiome – but sensitive to thermal stress – and some were not (and continued to be found in low abundances or weren’t even found, as members of the detectable microbiome). In any case, for all of these studies, there were some discussions that are in agreement with your results and discussion here (which is very positive as it shows consistent results over different coral probiotic applications). For all these studies, it seems that probiotics don’t need to be dominant or even persistent. In some cases, most probiotics won’t even be detected, as they only seem to trigger a beneficial microbiome restructuring/succession (e.g., Santoro et al., 2021). Overall, they do seem to guarantee niche protection and provide beneficial mechanisms that, collectively, protect corals against different impacts. These strategies may be slightly different when comparing disease and thermal stress, as the production of antibacterial compound seem to be crucial for SCTLD, for example, although they may all benefit from overall health improvement. Please adapt this paragraph to reflect that all studies developed so far seem to point to the same direction, that even if some of the probiotic players are

not dominant or persistent, they still seem to trigger beneficial outcomes, as this actually sounds like excellent news.

This paragraph was substantially restructured for better flow and focus (line 449-465).

Lines 328-322: I agree this is an incredible contribution, and maybe it should be also highlighted in the abstract, if possible.

Thank you, the abstract has been modified more to reflect this but we were limited by the word limit (abstract).

Figure 6: I am not sure I understand it, have you quantified it in non-inoculated corals, to evaluate the natural occurrence of these genes, as a control?

Samples were taken from each coral colony before inoculation as designated by “pre” on the x-axis in Figure 6.

Line 385: Did you keep healthy and sick corals under different conditions and feeding regimes? For how long? Did you acclimate them all under the same conditions and using the same system before starting the experiment?

No, the diseased corals were used within 72 h of collection because they were constantly losing tissue. None of the corals were fed during experiments to prevent further water fouling with the diseased corals. These points were clarified within this section (lines 554-555).

Line 490: Please add the number of replicates here.

The number of replicates were added to this sentence (line 666).

Line 508: When did you collect your samples for microbiome/ddPCR analyses? Just before inoculation?

The subsection describing “Treatment of diseased corals with McH1-7” did not include microbiome analyses. A separate experiment was set up dedicated to microbiome analysis. Those methods are described briefly under the subsection titled “Monitoring McH1-7 colonization over time through droplet digital PCR (ddPCR)” as well as in the supplemental material under the same subsection title. The time points for microbiome sampling were as follows:

“One fragment of each genotype was sacrificed at the following time points: before inoculation with McH1-7, and 1 day, 3 days, 7 days, 21 days, and 28 days post-inoculation for a total of 24 coral samples.”

Reviewer #2 (Remarks to the Author):

The paper by Ushijima describes the identification of a *Pseudoalteromonas* bacterial strain with novel potential use as a biological treatment for a coral disease. In my opinion, the research described in this paper is quite appealing as it is a good example of the potential use of microorganisms as biological pest control agents, with the additional difficulty to do so in aqueous media. However, as a reviewer I miss additional information about the state of the art in this field in a marine environment.

In addition, as a natural product chemist I believe that a deeper study of the chemical composition of the studied microorganism is necessary, particularly when the authors claimed in the title to have done it.

1.- The authors identified korormicin as one of the responsible agents of the bioactivity. Did they find any related molecule ? Many derivatives have been found from the same genera and may be present here as well.

Yes, subsequent experiments were conducted and various potential korormicin analogs were discovered and is described in Figure S11.

2.- Regarding chemical analysis, the identification of korormicin is trivial as it is a well known molecule. However, it is not easy to confirm this point by the reader. I could not find any figure in the manuscript and it is absolutely necessary to follow the tables in SI. I would recommend to include a table with NMR chemical differences between the herein reported values and the previously described ones (from synthesis).

We agree that the identification of a known antibiotic is not the major discovery of this paper, therefore, we feel like this additional figure would detract from the more important points of the manuscript, which are the effects of the bacterium on diseased corals and, to a lesser extent, the putative identification of the previously unknown korormicin biosynthetic gene cluster.

3.- TBP was also identified but I cannot find any experimental data in the SI section to confirm its identity.

Additional experiments and results for those experiments were added to the results section (line 237-239) and SI section to detail the identification of TBP.

In summary, I think the idea and discoveries behind this paper could be quite useful but I miss more scientific soundness from a chemical point of view. Although the reported research is well done, the identification of two-three well known compounds is not enough to claim the chemical characterization of a microorganism. Alternatively, if korormicin is the main responsible of the reported action I would make it clear although this would clearly lower the novelty of the discovery reported. Thus, taking into account that it seems difficult to use *Pseudoalteromonas* directly as a biological control agent shortly, I clearly miss a deeper chemical study of the bacterial extract or even a metabolomics study.

We agree with the points and therefore put further work into the further chemical characterization of Mch1-7. However, we believe the main focus should be more on the activity of Mch1-7 against SCTL

since this is the first description of any probiotic demonstrated to have an effect on a coral tissue loss disease.

Reviewer #3 (Remarks to the Author):

This manuscript describes a probiotic and prophylactic bacteria called McH1-7 for an ongoing coral disease outbreak – stony coral tissue loss disease. The authors use an interdisciplinary approach including microbiology, biochemistry, aquatic experiments, and ‘omics approaches to characterize the efficacy, antibacterial properties, and other characteristics of McH1-7. The results from the tank experiments are convincing, showing reduced mortality rates and tissue loss from SCTLD when McH1-7 is applied before and during infection. I also think the manuscript reads well, and I appreciated the detailed methods.

However, I wondered why the authors did not conduct a more thorough examination of the genome of McH1-7. The genome was characterized with antiSMASH, and then the other analyses were conducted through a targeted approach looking for specific genes. It seems relevant to conduct an entire functional analysis of the genome with KASS, KEGG, PFAM, or any other protein/functional annotation method. There may be other key genes/pathways that may bring to light the mechanisms of McH1-7.

Thank you for bringing this up. While we agree that a more comprehensive description of the genome is in order, there are two distinct reasons why we kept this brief. First, our intention is to publish a separate comparative genomics analysis to place the McH1-7 genome in the context of other marine *Pseudoalteromonas*. Second, with such an interdisciplinary study, our space within the manuscript is limited and many details have already been moved to the supplemental material. We chose therefore to focus in this paper on the pathways and genes that are linked to the production of antimicrobial compounds that presumably make McH1-7 an effective probiotic strain and that tie in with the chemical and physiological evidence we have provided. That said, we agree that some additional genome description is warranted and have added a table of COG category distribution (Table S6). In addition, the following lines were added to the subsection titled “Genomic analysis of McH1-7 identified genes for the antibacterial marinocine and chemical cue tetrabromopyrrole“:

“A thorough comparative genomic analysis of McH1-7 with other marine *Pseudoalteromonas* is ongoing. Here, we focus on the genes and biosynthetic gene clusters that putatively encode the antimicrobial compounds that presumably make McH1-7 an effective probiotic strain.”

As for the 16S rRNA results, the authors claim that there is no change to the microbiome by McH1-7 but also that McH1-7 was only detected in one sample at one time point. Since McH1-7 was only detected in one sample I would assume that there would be no obvious microbiome changes caused by McH1-7 and that it of course isn’t a dominant member if it can’t be detected. Based on these results it’s difficult to conclude that McH1-7 colonized the corals – perhaps it mostly colonized sick corals when added? I think the authors need to provide a deeper discussion on this discrepancy. I recognize that the authors detected the McH1-7 *Kor23* gene in the ddPCR assay in 5/24 samples, but this is still a low number of samples and the ddPCR values themselves are low, with not even one copy detected per ng. What does that mean in terms of the number of cells? Since McH1-7 is cultured, why not determine its detection threshold with this ddPCR assay.

The statements 1)“the authors claim that there is no change to the microbiome by McH1-7” and 2)“that McH1-7 was only detected in one sample at one time point” are both incorrect.

1) We did not claim that there is no change to the microbiome by McH1-7, rather we reported that the microbial communities were highly variable among coral fragments and addressed some of the reasons for this variation.

2) While we have been clear that McH1-7 was detected at very low levels through both 16S rRNA amplicons and ddPCR, we did not state that McH1-7 was only detected in one sample. Rather, McH1-7 was only detected in one sample using both methods. It is clear that both molecular methods were near their detection limits. However, the low detection of probiotic bacteria alongside apparent health benefits is consistent with the recent work of Santoro et al 2021 (Science Advances). The beneficial bacterial strains added by Santoro et al were all detected at less than 1% relative abundance.

Overall I think this was a well-done study, and I only have a few line-by-line comments below.

Line 32 Describe what is meant by “not fully characterized ...”

We agree this was a vague statement, and was changed to “...a waterborne disease with unknown etiology...” (line 44)

Line 43 Antibiotics was used to imply that this was a bacteria disease and not the other way around.

This is clarified in the sentence (line 55).

Line 147 What is the number of ORFs and GC content of the genome?

That information was added to the subsection titled “Genomic analysis of McH1-7 identified genes for the antibacterial marinocine and chemical cue tetrabromopyrrole”:

“The McH1-7 genome had 4,429 genes and a G+C content of 42.5%.” (lines 215-217)

Line 235 I think this needs to be its own section with a header

Agreed. The following subsection header was added: “Detection of McH1-7 in healthy corals over time after inoculation” (line 349)

Line 259 add space after Koromicin

Done.

Line 285 Let the reader know what method was used to detect this molecule.

This section was revised according to this suggestion.

Line 299 again remind the reader how this result was generated.

This section was revised according to this suggestion.

Line 314 - 317 This sounds like an introduction to a discussion and sounds out of place here. Perhaps split the discussion up into sections or rewrite.

Agreed. This paragraph was restructured for better flow and focus.

Line 325 How does this compare to probiotics used in other systems including corals? Is it typical that the probiotic isn't detected?

Yes, this is typical in studies that track the strains. Santoro et al 2021 (Science Advances) only detected 3 of 6 strains (and only at low levels of <1% relative abundance). Damjanovic et al 2019 (Frontiers in Microbiology) used a cocktail of 7 strains and 4 were detected at low levels (~1% relative abundance). This information has been added to the discussion.

Line 440 Which antibiotics were tested for the Kirby-Bauer tests?

The fractions from the extracts were used in the disc-diffusion assays, not specific antibiotics. The Kirby-Bauer test just refers to the assay protocol.

Line 461 Why was only the most non-polar fraction selected?

We were specifically looking for the presence of TBP in this section, which was predicted to be present in the non-polar fraction.

Line 556 I think it's helpful to add that these were healthy corals in the header of this section.

The header in the results, methods, and supplemental material were edited to reflect that the corals used were healthy.

In the supplementary it is difficult to provide comments since line numbers were not provided. But there is a paragraph that is completely underlined and I don't understand why.

The underlined paragraph has been fixed.

Fig 2 The text on the top portion of the figure (the cluster) is low resolution.

A higher resolution figure will be uploaded with the submission.

Fig 3 Remove the triangles from "I" since there is no key and there are no triangles in "K". In the legend add the percentage of tissue loss that is represented in A-H.

This figure and figure legend has been modified according to this suggestion.

Fig S9 add a Kb scale

Done

Fig S12 A. I think the data would be easier to interpret if each time point was plotted against the distance to the centroid. B. I think this should be plotted at a lower taxonomic level like genus especially if *Pseudoalteromonas* is apparent on the plot.

A panel showing the distance to centroid over time was added to this figure, which is now Figure S16. For the panel with stacked bar charts, using a lower taxonomic level would make the figure too crowded and unreadable – there are a total of 60 classes (shown), but there are 263 genera. In addition, *Pseudoalteromonas* ASVs were only found at very low levels throughout the samples and would not be apparent in the plot.

REVIEWERS' COMMENTS:

Reviewer #3 (Remarks to the Author):

Dear authors,

I feel that you have addressed all the points included in my report and after corrections the paper has clearly improved. From my point of view, the paper is now complete and doesn't need further changes to be published.

Final Revision Instructions

To the Author— Please review the editorial comments and requests below and confirm that changes have been made in the manuscript in the right-hand column. **This document must be uploaded** as a related manuscript file.

Please see our final file submission checklist for information about submitting your revised documents.

Files and General Policies	
Main manuscript file must be in Microsoft Word or LaTeX format. LaTeX and Tex article source files must be accompanied by the compiled PDF for reference. The bibliography must be submitted separately (as a .bib file) or contained within the .tex file.	The manuscript will be uploaded as a .docx file named “Mch1_7_paper_ComBio_Final.docx” with the bibliography contained within the text.
Each Figure must be provided as a separate file and must be supplied whole, with all panels included in a single document. Figures should be provided at a minimum resolution of 300 dpi at final size. Figure files must only contain images (please also leave out labels such as “Figure 1” etc). Figure captions must instead be included within the main manuscript file, grouped together at the end of the document.	Each of the 6 main figures will be uploaded as .tiff files and are each 300 dpi. The figure legends and the table is grouped at the end of the main document file “Mch1_7_paper_ComBio_Final.docx”
All figures, tables, and supplementary items must be cited in the manuscript and numbered in the order in which they appear.	All figures and supplementary figures are confirmed to be cited in the manuscript. They are grouped by figures, tables, supplementary figures and supplementary tables and ordered as they appear in the manuscript.

Tables in the main manuscript must be provided in an editable format and should be grouped together at the end of the main manuscript file.	The main table is at the end of the main document file “McH1_7_paper_ComBio_Final.docx”
Please check whether your manuscript contains third-party images, such as figures from the literature, stock photos, clip art or commercial satellite and map data. We strongly discourage the use or adaptation of previously published images, but if this is unavoidable, please request the necessary rights documentation to re-use such material from the relevant copyright holders and return this to us when you submit your revised manuscript. An appropriate permissions statement must be present in the relative figure caption for any third-party images.	No third-party images are included in this submission.
Please check that you have not copied any text directly from published work (even your own) without clear attribution, including one or more references. We run a plagiarism detection software and may need to request additional changes if we identify large blocks of identical text.	We have run the document through Grammarly and it didn't pick up anything for plagiarism.
An updated editorial policy checklist that verifies compliance with all required editorial policies must be completed and uploaded with the revised manuscript. All points on the policy checklist must be addressed; if needed, please revise your manuscript in response to these points. https://www.nature.com/documents/nr-editorial-policy-checklist.pdf. Please note that this form is a dynamic 'smart pdf' and must therefore be downloaded and completed in Adobe Reader. This file will not open in an internet browser.	This checklist had been completed and will be uploaded under the file named “nr-editorial-policy-checklist_UshijimaFinal.pdf”

The reporting summary will be published alongside your manuscript therefore it needs to accurately represent your work. In this case, please take a closer look at the reporting summary and make sure things are completed correctly. If an item does not apply, for example human participants, I need you to check the NA box next to that item. No section should be left blank. Also, please make sure to include your name and date at the top of the document. If you require a new Reporting Summary form, please download it here: https://www.nature.com/documents/nr-reporting-summary.pdf. Please note that this form is a dynamic 'smart pdf' and must therefore be downloaded and completed in Adobe Reader. This file will not open in an internet browser.	The reporting summary has been completed and uploaded under the file named "nr-reporting-summary_UshijimaFinal.pdf"
Your paper will be accompanied by a brief editor's summary when it is published on our homepage. Please approve the draft summary below or provide us with a suitably edited version (no more than 250 characters including spaces). A multidisciplinary approach identifies broad-spectrum antibacterial activity of Pseudoalteromonas sp. strain McH1-7 against stony coral tissue loss disease, which is threatening Caribbean coral reefs.	We approve of the summary.
ORCID Communications Biology is committed to improving transparency in authorship. As part of our efforts in this direction, we are now requesting that all authors identified as 'corresponding author' create and link their Open Researcher and Contributor Identifier (ORCID) with their account on the Manuscript Tracking System (MTS) prior to acceptance. ORCID helps the	This will be done upon upload.

scientific community achieve unambiguous attribution of all scholarly contributions. For more information please visit http://www.springernature.com/orcid. For all corresponding authors listed on the manuscript, please follow the instructions in the link below to link your ORCID to your account on our MTS before submitting the final version of the manuscript. If you do not yet have an ORCID you will be able to create one in minutes. https://www.springernature.com/gp/researchers/orcid/orcid-for-nature-research IMPORTANT: All authors identified as ‘corresponding author’ on the manuscript must follow these instructions. Non-corresponding authors do not have to link their ORCIDs but are encouraged to do so. Please note that it will not be possible to add/modify ORCIDs at proof. Thus, if they wish to have their ORCID added to the paper they must also follow the above procedure prior to acceptance. To support ORCID's aims, we only allow a single ORCID identifier to be attached to one account. If you have any issues attaching an ORCID identifier to your MTS account, please contact the Platform Support Helpdesk at http://platformsupport.nature.com/	
We regularly highlight papers published in Communications Biology on the journal’s Twitter account (@CommsBio). If you would like us to mention authors, institutions, or lab groups in these tweets, please provide the relevant twitter handles in the right-hand column.	@UshijimaLab, @PolypProtectors, @lebetimonas, @DrJMSneed, @UfYding
We would welcome the submission of material for the ‘Featured Image’ section on the Communications Biology home page. Images should relate to	One will be uploaded with the manuscript.

the content of your manuscript but need not be contained within the paper. Photographs and aesthetically interesting images are preferred; diagrams are generally not used. Suggestions should be uploaded as a Related Manuscript file. Please provide 1200x675-pixel RGB images. You will also need to submit a completed Image License to Publish. Unfortunately, we cannot promise that your suggestions will be used.	
Supplementary information	
Supplementary Information Format and referencing  ● Supplementary Figures, small Tables, and any supplementary text must be provided in a single PDF. Figures and their captions should be presented together.  ○ If you include a title page, please check that the title and author list matches the main manuscript. ● All Supplementary items must be referred to in the manuscript, and items must be mentioned in numerical order. Please do not include general references to “Supplementary Material”; instead refer to specific items. ● Additional files can be provided as Supplementary Data (Excel files, text files, .zip folders), Supplementary Movies, Supplementary Audio, or Supplementary Software (.zip folder) Supplementary Information files will be uploaded with the published article as they are submitted with the final version of your manuscript. Any highlighting or tracked changes should be removed from the file.	A single PDF will be uploaded containing all of the supplementary material.

Supplementary items must be cited in a consistent format. Names of items in the Supplementary file(s) must match those used in the main manuscript. We recommend using the following naming formats: Supplementary Figure 1, Supplementary Table 1, Supplementary Data 1, Supplementary Note 1, and Supplementary References. In this case, we would encourage you to incorporate the Supplementary Results and Methods in the main manuscript, however, if you keep these in Supplementary, please rename Supplementary Results to Supplementary Note.	The supplementary material has been renamed as a consistent format and Supplementary Results have been renamed to Supplementary Note.
Large tables and other data types: We strongly recommend depositing these to suitable repositories (such as Figshare, Dryad, or a data type-specific repository if one exists). Otherwise, these must be supplied as Supplementary Data files. Each file must be labelled as Supplementary Data 1, etc.	Large Data tables have been uploaded in Fishare and cited in the manuscript.
It's mandatory to provide access to the numerical source data for graphs and charts: We strongly recommend depositing these to suitable repositories (such as Figshare, Dryad, or a data type-specific repository if one exists). Otherwise, all source data underlying the graphs and charts presented in the main figures must be uploaded as Supplementary Data (in Excel or text format). Note that only the data used directly for generating the charts needs to be supplied.	They have been uploaded in Fishare and cited in the manuscript.

For any Supplementary Files such as those mentioned above that are not included your combined PDF (e.g. Supplementary Data, Movies, Audio, Software), please provide a title and description for each file here in the column to the right. For example: File name: Supplementary Data 1 Description: The source data behind the graphs in the paper	File Name: Chemical and genomic characterization of the first potential probiotic treatment for a coral disease threatening the Caribbean Description: Total tissue area remaining over time on stony coral tissue loss disease infected Montastraea cavernosa colonies. Coral fragments either received a direct treatment of McH1-7 or filtered seawater (control) or were tied to another M. cavernosa colony that was treated with McH1-7 or filtered seawater (control). Uploaded to figshare: https://figshare.com/s/18f60a82d908192c4014
Title Page	
Please ensure that the author list provided in our manuscript tracking system matches the author list in the main manuscript.	They all match.
Manuscript title Please ensure the title clearly describes the central finding of the paper. We recommend writing the title as a declarative statement of approximately 15 words or fewer.	We accept the recommended title by the editors.

Be sure to include any key species, protein names, or gene names to ensure optimal retrieval of the paper in database searches. The editors recommend the following title: “Chemical and genomic characterization of a potential probiotic treatment for stony coral tissue loss disease”	
Abstract The abstract should be accessible to non-specialists and avoid jargon and abbreviations. Please write the abstract in the present tense. We recommend structuring the abstract as follows: Opening statement explaining why this area of research is important. A sentence explaining the gap in knowledge that your research will address. Here we show (or an equivalent phrase), and then the major results and conclusions of the paper. Final sentence indicating any broader impacts and how this research will be used in the future.	The abstract has been modified to fit this structure.
Main text	
Format of the main text Please ensure your manuscript includes the following sections, presented in this order:  1. “Introduction”: The background and rationale for the work. The final paragraph should be a brief summary of the major results and conclusions. The results of the current study must only be discussed in 	The manuscript has been checked to fit this format.

this final paragraph. The Introduction should contain no references to figures or tables. Do not include subheadings.  2. “Results” or “Results and Discussion”. This should be split into subheaded sections; we recommend 1 subheading per main figure or table. Figures should not be embedded in the text but submitted separately.  a. Do not use more than 1 layer of subheadings. b. A “Conclusions” paragraph can be included only if the results and discussion are combined into a single section. 3. “Discussion” (optional), without subheadings. 4. Methods, which should be split into subheaded sections. Do not use more than 1 layer of subheadings. To improve readability, we recommend that the main text (Introduction, Results and Discussion) be limited to approximately 5000 words or fewer.	
Statistical reporting Wherever statistics have been derived (e.g. error bars, box plots, statistical significance) the legend needs to provide and define the n number (i.e. the sample size used to derive statistics) as a precise value (not a range), using the wording “n=X biologically independent samples/animals/independent experiments” etc. as applicable.	These are present.
Statistical representation Statistics such as error bars cannot be derived from $n < 3$ and must be removed from all such cases. We strongly discourage deriving statistics from technical replicates, and they should be removed from all such cases, unless	Nothing is derived from $n < 3$.

there is a clear scientific justification for why providing this information is important. Conflating technical and biological variability, e.g. by pooling technically replicate samples across independent experiments is strongly discouraged.	
Please include exact p-values where possible. We ask that you also include the name of the statistical test and the estimated effect size. If applicable, please also include the confidence interval.	These are included.
Avoid the use of the word “significant” unless referring the results of a statistical test.	Done.
Please check that all gene and mRNA names are in italics. Protein names should not be in italics. Please confirm that only official gene/protein symbols are used and that species names are in italics.	Done.
Language such as “new”, “novel”, “for the first time”, “unprecedented”, etc, should be avoided, or qualified with “to the best of our knowledge” or similar, because it often leads to unproductive controversy. Novelty should be made clear from the context. Please remove or qualify claims of “the first” in the title and on lines line 95, 285, 294, 382	This wording has been removed from the text accorind to the editors suggestions.
Display items	
Figure captions/legends Figures must have a title that will appear above the Figure and a legend that will appear below the Figure (see e.g. https://www.nature.com/articles/s42003-020-1059-1/figures/1)	The figures panels have been changed to lower case letters and their citation in the text have also been modified.

The Figure title must describe the Figure as a whole and must not contain reference to specific figure panels. The Figure legend must refer to and describe all panels. Abbreviations, symbols, colors, and shading present in the Figure must be defined. Please write out the symbols/colors in words (blue circles, red dashed line, etc.) within these definitions. All figure panels must be labelled using lower case letters (and cited as such in the text). Please refrain from referring to sections of figures as top/bottom/left/right/, etc. Please describe each panel using a letter.	
Axis and panel labels will be published as received. We recommend using a sans-serif font such as Arial or Helvetica.	
Data presentation in bar graphs and line graphs For all graphs depicting a single point value (e.g., mean) with error bars, you must add individual data points or convert the graph to a boxplot or dot-plot. You may wish to refer to this blog post about representing data distribution in plots (particularly for small datasets). We strongly encourage the same for plots with multiple time courses depicted. See the June 24, 2019 CommsBio editorial for more details about this policy. Example plots are shown here:	Individual data points were added to the plots or converted to a boxplot.

Examples of plots showing data distribution. Figure 2 from the editorial linked to above.

 Multiple line plot: After (converted to box plots)	
When choosing a color scheme please consider how it will display in black and white (if printed), and to users with color blindness. Please consider distinguishing data series using line patterns rather than colors, or using optimized color palettes such as those found at https://www.nature.com/articles/nmeth.1618. The use of colored axes and labels should be avoided. Please avoid the use of red/green color contrasts, as these may be difficult to interpret for colorblind readers.	Done.
Please define the error bars in each Figure and Supplementary Figure where they are used. One statement at the end of each Figure caption is sufficient if the error bars are equivalent throughout the Figure.	Done.
Tables in the main text Please check that your Tables comply with the following:	Done

 ● Do not include shading or colors. All Tables must contain black and white text only. ● Any bold/italic formatting must be either removed or defined clearly in a Table footnote. ● Where Tables contain images, each image should appear in its own cell in the absence of any text. ● All Tables must have a brief title. 	
Please pay close attention to our Digital Image Integrity Guidelines. Also ensure that you retain unprocessed data and metadata files after publication, ideally archiving data in perpetuity, as these may be requested during the peer review and production process or after publication if any issues arise.	Done
Methods	
Please ensure that all information present in the Reporting Summary is also in the manuscript. This information is usually most appropriate in the Methods section.	Done
We allow unlimited space for Methods. The Methods must contain sufficient detail such that the work could be repeated. It is preferable that all key methods be included in the main manuscript, rather than in the Supplementary Information. Please avoid use of “as described previously” or similar, and instead detail the specific methods used with appropriate attribution.	Done.
The Methods should include a separate section titled “Statistics and Reproducibility” with general information on how the statistical analyses of	A statistics and reproducibility section was added at the end of the methods section.

the data were conducted, and general information on the reproducibility of experiments (also those lacking statistical analysis), including the sample sizes and number of replicates and how replicates were defined.	
Data Policies	
Please add a Data Availability statement. The Data Availability statement must include:  ● Access details for deposited data, including repository name and unique data ID. ● How source data can be obtained. ● A statement that all other data are available from the corresponding author (or other sources, as applicable) on reasonable request. Note that ‘available upon request’ is only appropriate if immediate data access has not been mandated by our policies or by the editors. See here for more information about formatting your Data Availability Statement: http://www.springernature.com/gp/authors/research-data-policy/data-availability-statements/12330880	This has been added to the manuscript.
Mandatory deposition of raw and processed data is required for:  ● All sequencing data (DNA, RNA, protein) ● Novel human genetic polymorphisms (e.g., dbSNP) ● Linked genotype and phenotype data (e.g., dbGaP for human data) ● GWAS summary statistics or polygenic risk scores ● Novel macromolecular structure ● Gene expression microarray data (must be MIAME compliant) ● Crystallographic data for small molecules ● Mass spectrometry-based proteomics data 	All data has been deposited.

For more information on mandatory data deposition policies at the Nature Portfolio, please visit http://www.nature.com/authors/policies/availability.html#data For an up-to-date list of approved repositories for each mandatory data type, please visit https://www.springernature.com/gp/authors/research-data-policy/repositories/12327124. Accession code(s) for deposited data must be provided in the Data Availability statement in the final version of the paper. Failure to do so will delay publication. Please ensure data are available prior to publication.	
Communications Biology has a strong preference for all data to be deposited in an approved repository. In some cases, data deposition may be required by the editor. We recommend the following data repositories: ● GenBank (all DNA sequence data)● NHGRI-EBI GWAS Catalog (GWAS summary statistics)● PGS Catalog (polygenic risk scores)● Gene Expression Omnibus (Microarray or RNA sequencing data)● Sequence Read Archive (WGS or WES data)● Protein Data Bank (protein structural data)● OSF (neuroimaging raw data and EEG/EMG/MEG raw data)● Neurovault (unthresholded statistical maps, parcellations, and atlases produced by MRI and PET studies)● Image Data Resource (microscopy data)	All data has been deposited.

 ● PRIDE (proteomics data) Data types without a specific repository can be deposited in a generalist repository, such as figshare or Dryad. For an up-to-date list of approved repositories, please visit https://www.springernature.com/gp/authors/research-data-policy/repositories/12327124.	
Data citation Please cite datasets stored in external repositories in the main reference list. For previously published datasets, we ask authors to cite both the related research articles and the datasets themselves. For more information on how to cite datasets in submitted manuscripts, please see our data availability statements and data citations policy.	Done
Please deposit your newly generated plasmids in a community repository (eg, Addgene). Include the ID numbers in the Data Availability statement.	N/A for this publication.
End Notes	
Please check that your bibliography complies with the following:  ● Your bibliography should start with the heading "References". The references must be numbered in the order of appearance in the text, then tables, then figures. ● Any in-text citations to references (e.g. "Gupta et al. show...") should be followed by their corresponding reference citation number from the reference list. 	Done

● Manuscript citations must include journal title, article title, volume number, page or article number or DOI, and year of publication.● No publication can be present more than once in the reference list.● No footnotes are permitted in the references or elsewhere. Text should be incorporated into the main text, the Methods section, or the Supplementary Information instead.● Websites should only be listed in the references if they are in common use or curated.● Where possible, preprints in the reference list should be updated with details of the published, peer-reviewed paper.● Citations should be formatted in the text using superscript numbers.	
Please check that your 'Author Contributions' section individually lists the specific contribution of each author to the work. Each author must be referred to by name or initials. Where multiple authors possess identical initials, they must be clearly disambiguated from one another. See our author contributions policy for further information: https://www.nature.com/nature-research/editorial-policies/authorship#author-contribution-statements	Done